# The induction of peripheral trained immunity in the pancreas incites anti-tumor activity to control pancreatic cancer progression

Anne E. Geller[1,2], Rejeena Shrestha[1,2], Matthew R. Woeste [1,2,3], Haixun Guo[4], Xiaoling Hu[2], Chuanlin Ding[2], Kalina Andreeva[5,6], Julia H. Chariker[5,6], Mingqian Zhou[2], David Tieri[7], Corey T. Watson [7], Robert A. Mitchell [2], Huang-ge Zhang[1], Yan Li [3], Robert C. G. Martin II[3], Eric C. Rouchka [6,7] & Jun Yan [1,2 ✉]

Despite the remarkable success of immunotherapy in many types of cancer, pancreatic ductal adenocarcinoma has yet to benefit. Innate immune cells are critical to anti-tumor immuno-surveillance and recent studies have revealed that these populations possess a form of memory, termed trained innate immunity, which occurs through transcriptomic, epigenetic, and metabolic reprograming. Here we demonstrate that yeast-derived particulate β-glucan, an inducer of trained immunity, traffics to the pancreas, which causes a CCR2-dependent influx of monocytes/macrophages to the pancreas that display features of trained immunity. These cells can be activated upon exposure to tumor cells and tumor-derived factors, and show enhanced cytotoxicity against pancreatic tumor cells. In orthotopic models of pancreatic ductal adenocarcinoma, β-glucan treated mice show significantly reduced tumor burden and prolonged survival, which is further enhanced when combined with immunotherapy. These findings characterize the dynamic mechanisms and localization of peripheral trained immunity and identify an application of trained immunity to cancer.

[1] Department of Microbiology and Immunology, University of Louisville, Louisville, KY, USA. [2] Division of Immunotherapy, The Hiram C. Polk, Jr., MD Department of Surgery, Immuno-Oncology Program, Brown Cancer Center, University of Louisville, Louisville, KY, USA. [3] Division of Surgical Oncology, The Hiram C. Polk, Jr., MD Department of Surgery, University of Louisville, Louisville, KY, USA. [4] Department of Radiology, University of Louisville, Louisville, KY, USA. [5] Department of Anatomical Sciences and Neurobiology, University of Louisville, Louisville, KY, USA. [6] Kentucky Biomedical Research Infrastructure Network Bioinformatics Core, University of Louisville, Louisville, Kentucky, USA. [7] Department of Biochemistry and Molecular Genetics, University of Louisville, Louisville, KY, USA. ✉email: jun.yan@louisville.edu

The diagnosis of pancreatic ductal adenocarcinoma (PDAC) is a devastating one, with only 10% of patients surviving the past 5 years[1]. Although the survival rate since 2014 has increased from 6 to 10%, pancreatic cancer remains refractory to the majority of currently available therapeutics. In addition, as the demographics of the United States shift, it is projected that pancreatic cancer will become the second leading cause of cancer-related mortality by 2030 and thus presents a significant future challenge for clinicians[2]. Pancreatic cancer is particularly lethal due to the fact that in early stages there are seldom clinical symptoms, which results in 75–80% of patients being diagnosed with advanced, non-resectable disease[3,4]. Even in patients who are eligible for resection, the 5-year survival rate is only 20–25%[4]. Furthermore, pancreatic cancer has shown little responsiveness to immunotherapies which have shown remarkable effects in other solid tumors[5–9]. The Phase I and II clinical trials using CTLA-4 and PD-1 inhibitors both alone and in combination have been deemed ineffective for the treatment of PDAC, which is likely explained by the non-immunogenic nature of PDAC[10–12].

A major challenge to the successful application of immunotherapy in PDAC is overcoming the immunosuppressive pancreatic tumor microenvironment (TME). PDAC is characterized by a dense pro-tumorigenic desmoplastic stroma, an abundance of immunosuppressive cell subsets within this stroma such as tumor-associated macrophages (TAMs), regulatory T-cells (T-regs), and myeloid-derived suppressor cells (MDSCs), a dearth of activated anti-tumor immune cells, and hypo-vascularity that lends to a hypoxic microenvironment[13–16]. Together these conditions make it exceptionally challenging to effectively deliver immunotherapies to the pancreas and for these therapeutics to successfully activate anti-tumor immune responses if they do arrive there. Therefore, novel therapeutics are desperately needed that can not only specifically target the pancreas, but that can also infiltrate the dense desmoplastic stroma, and that are capable of inciting robust antit-umor immune responses despite the immunosuppressive TME.

Trained innate immunity is an evolutionarily ancient program of immunological memory that has recently come under in-depth scientific investigation. Trained innate immune cells have been shown to undergo transcriptomic, epigenetic, and metabolic reprogramming upon exposure to specific initial stimuli, and when these innate immune cells are reexposed to a secondary heterologous stimulus, they are "trained" to be more responsive to that stimulus which manifests in an enhanced inflammatory response[17–19]. There are many biologics that are able to induce trained immunity including the Bacillus Calmette-Guérin (BCG) vaccine and the natural compound β-glucan. Though most studies regarding trained immunity focus on pathogens such as bacteria and viruses as a secondary stimulus, new studies suggest that tumor cells may also reactivate trained immune cells. All research to date has utilized subcutaneous models of cancer, therefore it is not known whether the presence of trained innate immune cells may invoke anti-tumor effects on tumors within solid organs, such as pancreatic cancer[20,21].

In this study, we unexpectedly discovered that intraperitoneally (IP) injected yeast-derived particulate β-glucan traffics in large proportion to the pancreas. The direct trafficking of β-glucan to the pancreas highlights a previously uncharacterized pathway for the induction of peripheral trained immunity. Additionally, we tested the hypothesis that this trafficking might result in the induction of anti-tumor immunity against pancreatic cancer. Given the failure of the majority of therapeutics and immunotherapies alone and in combination to treat PDAC and the difficulty of targeting these therapeutics to the pancreas, our findings collectively provide a potential breakthrough in not only targeting the pancreas directly but also in recruiting anti-tumor, innate immune cells to the immunologically cold PDAC TME.

## Results

**Yeast-derived particulate β-glucan preferentially traffics to the pancreas.** It has been well documented that the administration of β-glucans from bacteria and fungi results in an increase in the numbers and frequencies of hematopoietic progenitors and multipotent progenitors that are biased towards the myeloid lineage, which ultimately functions as an important step in the induction of central and peripheral trained immunity[22]. However, it has not yet been shown where exactly β-glucan traffics to after administration in order to have these effects. In these studies, whole β-Glucan particles (WGP) derived from *Saccharomyces cerevisiae* were used, which are 2–4 micron hollow yeast cells made of highly concentrated (1,3) β-glucans. The detailed characterization of WGP β-glucan is described in one of our other studies. Unlike other forms of β-glucan used in trained immunity, this formulation is unique in that it is particulate and requires active phagocytosis to exert its effects. To assess the trafficking of this particulate β-glucan, WGP was tagged with (5-(4,6-Dichlorotriazinyl) Aminofluorescein) (DTAF), and injected IP into wild-type (WT) C57BL/6 mice. Three days following IP administration (3-day WGP), the lung, spleen, inguinal and mesenteric lymph nodes, peritoneal cavity cells, and pancreas were harvested, washed with ice-cold PBS to wash away DTAF-WGP from the peritoneum that may have been adhered to the surface of the organ, and the presence of DTAF-WGP in each organ was assessed by flow cytometry. While there was some trafficking of the DTAF-WGP to the spleen, mesenteric lymph nodes, and residual DTAF-WGP in the peritoneal cavity, the pancreas showed a striking and unexpected presence of the DTAF-WGP (Fig. 1a). To further assess this trafficking and to ensure that the DTAF label was not involved in the trafficking mechanism, WGP was radiolabeled with [89]Zr and injected IP (Fig. 1b) or incubated with peritoneal macrophages that were then injected IP (Fig. 1c). Mice were first imaged using a PET/CT scan 48 h following injection and green circles are used to indicate the observed preferential accumulation of the [89]Zr-WGP in the pancreas. A necroscopy was then performed and the radioactive signature of each organ was measured. In accordance with the flow cytometry data, [89]Zr-WGP trafficked in large quantities to the pancreas and was found in lower levels in the spleen, liver, and intestinal system. Peritoneal macrophages that were cultured with [89]Zr-WGP and then injected IP had similar though slightly more diversified trafficking than the pure [89]Zr-WGP, and also accumulated most prominently in the pancreas (Fig. 1d). Trafficking of peritoneal macrophages loaded with WGP to the pancreas indicates that both naked WGP as well as macrophages that have phagocytosed WGP display tropism to the pancreas.

In an effort to assess the role that the known receptor of WGP played in the trafficking of WGP, mice lacking the C-type lectin receptor, Dectin-1, (Dectin-1[−/−] mice) were injected IP with DTAF-WGP. As compared to WT mice, Dectin-1[−/−] mice showed a fivefold decrease in the amount of WGP that trafficked to the pancreas, as assessed by flow cytometry (Fig. 1d). [89]Zr-WGP was also injected IP into WT mice and Dectin-1[−/−] mice. As compared to WT animals, there was significantly less trafficking of [89]Zr-WGP to the pancreas of Dectin-1[−/−] mice (Fig. 1e). To ensure that this process was in fact β-glucan specific, a polystyrene-based latex 3 μm fluorescent particle, the same size as a WGP particle, was injected IP and was not found to accumulate in the pancreas (Supplementary Fig. 1a). Together, these data highlight a previously uncharacterized dectin-1 dependent tropism in which β-glucan traffics to the pancreas.

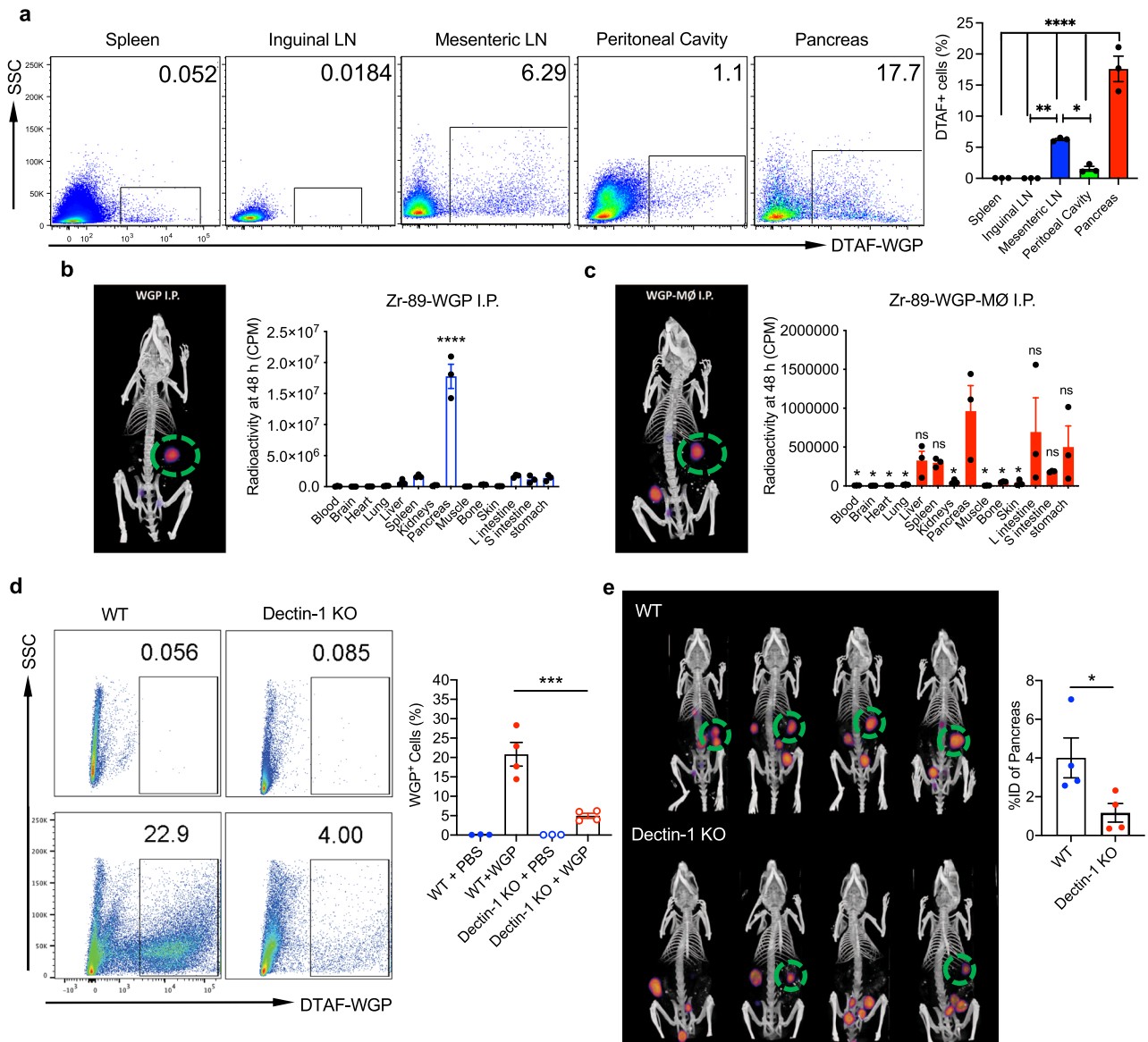

**Fig. 1 Particulate β-glucan traffics to the pancreas in a dectin-1 dependent manner. a** DTAF-WGP was injected I.P. and 3 days later different tissues ($n = 3$) were harvested and assessed for the presence of the DTAF-WGP by flow cytometry. Representative dot plots and summarized data are shown. *$p = 0.032$, **$p = 0.0057$, ****$p < 0.0001$. **b** WGP was labeled with [89]Zr-WGP or **c** peritoneal macrophages were incubated with [89]Zr-WGP and washed, followed by I.P. injection. PET/CT imaging displays the trafficking of the [89]Zr-WGP after 48 h. Organs were individually assessed for radioactivity following a necroscopy using a gamma counter. In **c**, significance is reported as compared to the pancreas ($n = 3$). ****$p < 0.0001$. **d** Dectin-1[−/−] mice or WT mice were injected with DTAF-WGP and the accumulation of DTAF-WGP in the pancreas was assessed by flow cytometry (WT PBS $n = 3$, WT WGP $n = 4$, KO + PBS $n = 3$, KO + WGP $n = 4$). **e** Dectin-1[−/−] mice or WT mice ($n = 4$) were injected with [89]Zr-WGP and 48 h later a PET/CT was used to assess the amount in the pancreas. The signal was quantified by reporting the % of injected dose (%ID) in the pancreas. *$p = 0.046$. A one-way ANOVA with Tukey's multiple comparisons was used in **a**–**d**, and an unpaired, two-tailed student's $t$-test was used in **e**. Data were represented as mean ± SEM. ns not significant; Each sample represents a biologically independent animal obtained over a single independent experiment which was repeated at least twice for verification of results.

**β-Glucan that traffics to the pancreas incites an influx of innate immune cells that show a phenotype of trained immunity.** After discovering that β-glucan displays tropism toward the pancreas, it was also observed that the immune landscape of the pancreas was significantly altered following WGP treatment. The arrival of WGP to the pancreas was accompanied by a distinct influx of CD11b[+] myeloid cells to the pancreas by day 3, some of which had phagocytosed WGP (Fig. 2a). This finding led us to examine how overall immune populations of the pancreas are impacted following IP WGP treatment. As it has been previously observed that the immune changes associated with trained

immunity in the BM are most pronounced 1 week following exposure to β-glucan[22], the immune profile of the pancreas was examined in mice treated with a 3 μm polystyrene microparticle or WGP 7 days after administration. First, it was observed that there was about a sevenfold increase in the overall absolute number of CD45[+] immune cells in the pancreas after WGP treatment (Fig. 2b). The 3 μm polystyrene microparticle control did not have such effects (Supplementary Fig. 1b), indicating that this immune cell influx is WGP specific. In addition, there was an observed increase in the percent of CD45[+] immune cells (Supplementary Fig. 1c). To further classify which cell types were

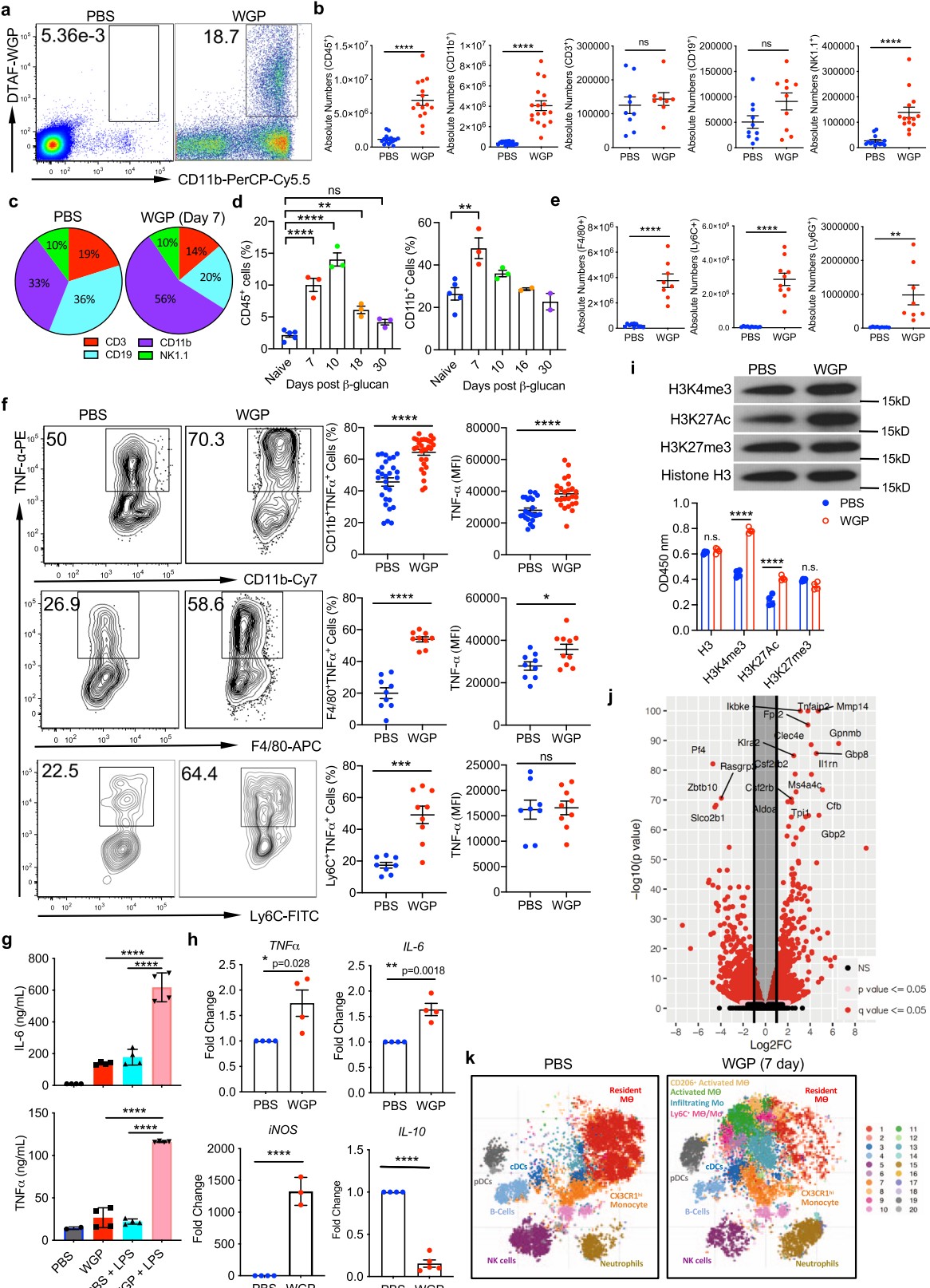

responsible for this expansion of the CD45$^+$ immune population, the absolute number (Fig. 2b) and relative percent (Supplementary Fig. 1c) of myeloid cells (CD11b$^+$), T-cells (CD3$^+$), B-cells (CD19$^+$), and NK cells (NK1.1$^+$) were evaluated. The cumulative changes in the pancreas are represented by pie charts which demonstrate that the expansion of the myeloid compartment is

responsible for the relative decrease in the percentage of other cell populations and the overall increase in the CD45$^+$ population following WGP treatment (Fig. 2c). We next investigated whether this influx of CD11b$^+$ immune cells was transient, so the percent of overall CD45$^+$ cells and CD11b$^+$ cells in the pancreas following IP injection was measured at 7,10, 18, and 30 days.

**Fig. 2 β-glucan stimulates an influx of trained myeloid cells into the pancreas. a** Percent of CD45[+]CD11b[+]DTAF[+] cells in the pancreas 3 days after WT mice received I.P. DTAF-WGP. **b** Seven days after PBS or WGP injection, absolute numbers of CD45[+] ($n = 15$), CD11b[+] ($n = 18$), CD3[+] ($n = 8$), CD19[+] ($n = 10$), and NK1.1[+] ($n = 14$) cells are shown. ****$p < 0.0001$. **c** Pie charts representing frequency changes of major immune cell populations after WGP treatment. **d** WT mice were injected with PBS ($n = 5$) or WGP ($n = 5$) and the percent of CD45[+] and CD45[+]CD11b[+] cells were measured 7, 10, 18, and 30 days later ($n = 3$). **$p = 0.0034$, ****$p < 0.0001$ (CD45), **$p = 0.0023$ (CD11b). **e** Absolute numbers of F4/80[+] ($n = 10$), Ly6C[+] ($n = 10$), and Ly6G[+] ($n = 8$) within the CD11b[+] population. **$p = 0.003$, ****$p < 0.0001$. **f** Seven days after IP PBS or WGP the pancreas was restimulated with LPS. TNFα production in CD11b[+] ($n = 28$), CD11b+F4/80[+] ($n = 9$), and CD11b+Ly6C[+] ($n = 8$) cells was measured. *$p = 0.024$, ***$p = 0.0001$, ****$p < 0.0001$. **g** Seven days after PBS or WGP, the CD45[+]CD11b[+] population was enriched ($n = 4$). Cells were restimulated with or without LPS for 24 h. TNFα and IL-6 production was measured using ELISA. ****$p < 0.0001$. **h** Pancreatic CD11b[+] cells from PBS and WGP-trained mice were sorted. RT-qPCR was done to quantify TNFα ($n = 4$), IL-6 ($n = 4$), iNOS (PBS $n = 4$, WGP $n = 3$), and IL-10 (PBS $n = 4$, WGP $n = 5$) mRNA expression levels. *$p = 0.028$, **$p = 0.0018$, ****$p < 0.0001$. **i** CD11b[+] cells from PBS or WGP-injected mice (24 h) were sorted ($n = 4$). Histones were isolated and subjected to western blot analysis (top) or ELISA (bottom) for H3K4me3, H3K27Ac, H3K27me3, and total H3. ****$p < 0.0001$. **j** Seven days after IP WGP or PBS, pancreatic CD45[+]CD1b[+] populations were sorted. RNA-Seq analysis was performed (PBS $n = 3$, WGP $n = 3$). The distribution of $p$ values ($-\log_{10}(p$ value)) and fold changes ($\log_2$ FC) of differentially expressed genes are represented. **k** t-SNE plots of the CD11b[+] population in mice trained with PBS or WGP 7 days prior and analyzed with CyTOF (PBS $n = 3$, WGP $n = 3$). Unpaired, two-tailed student's $t$-tests were used in **b**, **e**, **f**, and **h**. A one-way ANOVA with multiple comparisons was used in **d**, **g**, and **i**. Data represented as mean ± SEM. ns not significant. Each sample represents a biologically independent animal obtained over a single independent experiment.

While the percent of CD11b[+] cells peaked at day 7, the percent of CD45[+] cells appeared to peak at day 10 which may indicate that following the influx of CD11b[+] cells to the pancreas, an influx of other immune cell types may follow. The percent of immune cells in the pancreas decreased to basal levels by day 30, demonstrating that this influx is transient (Fig. 2d). Within the myeloid compartment, we observed the absolute number (Fig. 2e) and percent (Supplementary Fig. 1c) of macrophages (F4/80[+]), monocytes (Ly6C[+]), and neutrophils (Ly6G[+]) were all increased. The influx of immune cells into the pancreas was also shown to be dependent on the dose of WGP injected, where higher doses of WGP were associated with increased influx of overall CD45[+]CD11b[+] myeloid cells and CD45[+]CD11b[+]F4/80[+] macrophages (Supplementary Fig. 1d). As pancreatitis and the destruction of pancreatic islets are associated with immune cell infiltration of the pancreas, H + E of the pancreas was performed to assess the integrity of acini 7 days after WGP administration (Supplementary Fig. 1e). Serum amylase, a diagnostic marker of pancreatitis, was also measured in WGP vs PBS-treated mice 7 days following injection (Supplementary Fig. 1f). Neither the islets nor the serum amylase was adversely impacted by WGP treatment and the observed immune influx, indicating that the immune cell influx in this mechanism does not cause pancreatic destruction.

To investigate whether the cells found to infiltrate the mouse pancreas following WGP treatment displayed a phenotype of trained immunity, we used a standard training protocol in which mice were treated with WGP or PBS and 7 days later the pancreas were harvested and then restimulated with LPS. TNF-α production has been used as a surrogate marker to evaluate the trained response. Overall CD11b[+] myeloid cells, CD11b+F4/80[+] macrophages, and CD11b+Ly6C[+] monocytes were all shown to produce more TNF-α due to prior exposure to WGP, as assessed by the percent of TNF-α[+] cells and the MFI (Fig. 2f). To further assess these findings, CD11b[+] cells were sorted from the pancreas of these mice and restimulated ex vivo with LPS, and the TNF-α and IL-6 levels in the supernatants were measured by ELISA. As compared to CD11b[+] cells from PBS mice, cells from WGP-trained mice that were restimulated with LPS produced significantly more TNF-α and IL-6 (Fig. 2g), signifying that WGP-treated pancreatic CD11b[+] cells were trained. These results were further confirmed using RT-PCR, where pancreatic CD11b[+] cells sorted from PBS or WGP-trained mice were shown to produce more TNF-α, IL-6, iNOS, and less IL-10 due to WGP treatment (Fig. 2h). Since trained immunity is associated with metabolic and epigenetic reprogramming[23], we measured lactate levels to examine whether glycolysis is involved. Indeed, WGP-

trained CD11b[+] cells secreted more lactate compared to untrained cells (Supplementary Fig. 1g). We also measured histone modifications of WGP-trained CD11b[+] cells. As shown in Fig. 2I, the expression levels of H3K4me3 and H3K27Ac were significantly enhanced in WGP-trained CD11b[+] cells. This was revealed by both western blot analysis (Fig. 2i, top) and ELISA quantitative assay (Fig. 2i, bottom). Together these findings indicate that not only does WGP incite a trained phenotype in the cells that traffic into the pancreas, but that they also undergo metabolic and epigenetic reprogramming.

RNA sequencing (RNA-Seq) was then performed on FACS sorted CD45[+]CD11b[+] cells 7 days following PBS or WGP administration to obtain an unbiased and comprehensive characterization of myeloid populations in the pancreas. A total of 1459 differentially expressed genes (DEGs) were discovered, with 661 upregulated and 798 genes downregulated in the WGP-trained setting (Fig. 2j). Gene set enrichment analyses (GSEA) further confirmed previously investigated upregulations in pathways related to TNF-α and, IL-6 production (Supplementary Fig. 2a). In accordance with the observed influx of immune cells, the DEGs related to leukocyte chemotaxis, leukocyte migration (Supplementary Fig. 2a), monocyte chemotaxis, macrophage migration, and myeloid leukocyte migration (Supplementary Fig. 2b) were all significantly enriched in the WGP group. Additionally, reactome pathways CLEC7A dectin-1 signaling and C-type lectin receptor pathways were significantly enriched, further corroborating the Dectin-1 involvement in this process (Supplementary Fig. 2c).

Considering that the myeloid population was observed to be responsible for the expansion of the CD45[+] population and that myeloid cells are a diverse population, CyTOF analysis was performed on pancreata from mice treated with PBS or WGP 7 days prior. Cell populations within the pancreas were identified and compared between treatment groups; t-SNE plots indicated that following WGP treatment there was a relative decrease in the resident macrophage population which highly expresses M2 markers such as CD206. We also observed an appearance of several myeloid populations in the pancreas which were defined as Ly6C[+] macrophage-derived monocytes, infiltrating inflammatory monocytes, activated macrophages, and CD206[+] activated macrophages (Fig. 2k and Supplementary Fig. 2d). viSNE plots gated on the CD11b[+] population showed a distinct expansion of monocytes and macrophages, with additional increases of the dendritic cell (DCs), neutrophil, and NK populations after WGP treatment (Supplementary Fig. 2e). Additionally, these analyses highlight that following WGP training and LPS restimulation,

TNFα was primarily produced by macrophages and monocytes, IL-6 and iNOS were primarily produced by macrophages, Granzyme-B was produced by NK cells and macrophages, and IFNγ was principally produced by neutrophils (Supplementary Fig. 2f). This reveals that while many cell populations are changed due to WGP treatment, the primary cells trained by WGP, as assessed by increased TNFα expression, are macrophages and monocytes and that there is not a single cell type responsible for the phenotype of the trained immunity that was observed.

To assess whether this phenomenon of myeloid cell influx and activation was dependent upon other immune populations or driven entirely by the myeloid cells themselves, we depleted specific immune populations and observed the trained phenotype. Anti-CD4 and anti-CD8 mAbs were used alone and in combination to deplete T-cells, the efficiency of depletion was assessed (Supplementary Fig. 3a), and the trained phenotype of myeloid cells was observed (Supplementary Fig. 3b, c). NK cells were also depleted using mAb PK136, the depletion efficiency was evaluated (Supplementary Fig. 3d) and the trained phenotype was observed (Supplementary Fig. 3e, f). In the absence of T-cells and NK cells, myeloid cells were still observed to be trained by WGP. The trained phenotype was also assessed in NSG mice which lack B-cells, T-cells, and NK cells (Supplementary Fig. 3g, h). These mice still showed a clear phenotype of trained immunity. Finally, we depleted neutrophils using Ly6G mAb. Despite the depletion of granulocytes, we still observed WGP-induced training in the CD11b$^+$ myeloid compartment (Supplementary Fig. 3i–k). Taken together, our data support that the infiltrating myeloid cells in the pancreas are trained by β-glucan without the assistance or influence of adaptive immune populations, NK cells, or granulocytes.

**Single-cell RNA sequencing reveals specific populations of pro-inflammatory macrophage/monocytes that traffic to the pancreas upon β-glucan treatment.** As our CyTOF analyses have revealed the appearance of several previously uncharacterized populations to the pancreas following WGP treatment, to gain a more in-depth understanding of the cell populations present in the pancreas, a single-cell RNA sequencing (scRNA-Seq) was performed on sorted CD45$^+$ cells from PBS-treated mice and mice treated with WGP three (3-day WGP) and seven days prior (7-day WGP). Two-dimensional UMAP representation of 11,132 cells aggregated from three samples with clusters resulting from k-nearest neighbors and Louvain algorithms partitioned into 19 distinct clusters (Fig. 3a, b). The relative frequency of each cluster within each experimental group was assessed. Here, significant increases in the frequency of clusters 3,4, and 10 were observed by day 7 after WGP treatment, along with a near disappearance of cluster 5 (Fig. 3c). The relative frequencies of several other populations were also shown to decrease over time. However, this apparent trend is likely due to the relative increase in the frequency of other populations. Overall, clusters 3,4,10, and 5 appear most significantly altered due to WGP treatment (Fig. 3d).

The populations noted previously to be most significantly altered over time due to WGP treatment were part of the myeloid compartment, as identified by *CSF1R* expression (Fig. 3d, e). As these data along with previous findings suggest that myeloid populations drive the immune changes associated with WGP treatment, the dynamic *CSF1R*$^+$ clusters 3,4,10, and 5 were investigated in more detail as shown by violin plots (Fig. 3f) and dot plots representing the top 12 marker genes by average Z-score for each cluster (Fig. 3g). Cluster 5, which was present in PBS mice but virtually disappeared after WGP treatment was identified as resident macrophages[24,25]. This cluster also notably expressed *MAF* and *APO* which are both related to M2 macrophage polarization[26–28]. Cluster 19 was largely unchanged

over time so was not described in more detail, but exhibited an expression profile similar to cluster 5, and thus also represents a subset of tissue-resident macrophages. Cluster 10, the Ly6C$^{lo}$ macrophage population, also had a similar expression profile to resident macrophage cluster 5, and notable expression signatures of *ARG1* and *FABP*. We hypothesize this population to be repolarized resident macrophages given the lack of Ly6C expression, similarity to the resident macrophage population, and the disappearance of the resident population following WGP treatment. Clusters 3 and 4 shared general phenotypic characterization as Ly6C$^{hi}$ infiltrating monocytes/macrophages. Cluster 4 had a notable expression of *Chil3* and *Plac8*, which together have been identified by another group to identify Ly6C$^{hi}$ infiltrating macrophages[29]. Finally, cluster 3, which was absent in naïve mice and showed the greatest increase in relative frequency among all clusters following WGP treatment expressed substantial signatures of *TNFAIP2*, *IL1B*, *SOD2*, and *PRDX5*, indicating a strongly pro-inflammatory phenotype. Cluster 3 was thus identified as Ly6C$^{Hi}$ inflammatory infiltrating monocytes/macrophages. Interestingly, within the myeloid subset, the populations shown to increase due to WGP treatment are the only populations to express *TNFAIP2*, indicating that the cells entering the pancreas are likely trained myeloid cells.

To further characterize the activation status of each of the four dynamic myeloid clusters, dot plots were constructed to show the relative expression of genes associated with a pro-inflammatory status and an anti-inflammatory status (Fig. 3h). Clusters 3 and 4, the infiltrating monocytes/macrophages, were shown to be pro-inflammatory, the resident macrophages were anti-inflammatory, and the macrophages in cluster 10 showed an intermediate phenotype. In agreement with previous CyTOF and flow cytometry data, this scRNA-Seq data further characterizes these newly identified myeloid cells in the pancreas to be a heterogeneous population of transformed and repolarized Ly6C$^{Lo}$ resident macrophages and pro-inflammatory Ly6C$^{Hi}$ infiltrating monocyte-derived macrophages that express signatures of trained immunity.

**The WGP-driven influx and training of myeloid cells in the pancreas is CCR2-dependent and occurs as early as 24 h post-WGP treatment.** We next examined what mechanisms were responsible for this influx of cells into the pancreas. RNA-Seq data was used to characterize chemokines and cytokines whose expression was significantly upregulated upon WGP treatment (Fig. 4a). While several chemokines and cytokines were upregulated, our observation of macrophage and monocyte influx into the pancreas piqued a specific interest in CCR2 due to its involvement in monocyte and macrophage recruitment and in monocyte egress from the bone marrow[30–32]. CyTOF data also showed a prominent increase in CCR2 positive cells after WGP treatment (Fig. 4b) and scRNA-Seq showed a distinct expression of CCR2 in clusters 3 and 4, which were the two populations that showed the most distinct phenotypes of trained immunity (Fig. 4c). Additionally, 24 h post-WGP treatment, whole pancreatic lysates showed a 30-fold increase in CCL2, which is the ligand for CCR2 and is involved in mediating monocyte chemotaxis (Fig. 4d).

RNA-Seq data had shown a clear signature of immune cell recruitment and trafficking, and these data had also indicated that WGP upregulated proliferation of leukocytes and mononuclear cells (Fig. 4e). To investigate whether the CCR2$^+$ myeloid cells were proliferating once they reached the pancreas, the percent of CCR2$^+$ cells expressing Ki67 was assessed in PBS and WGP-treated mice. Following WGP treatment there was an increase in overall proliferating cells (Fig. 4f) and a large percent of these

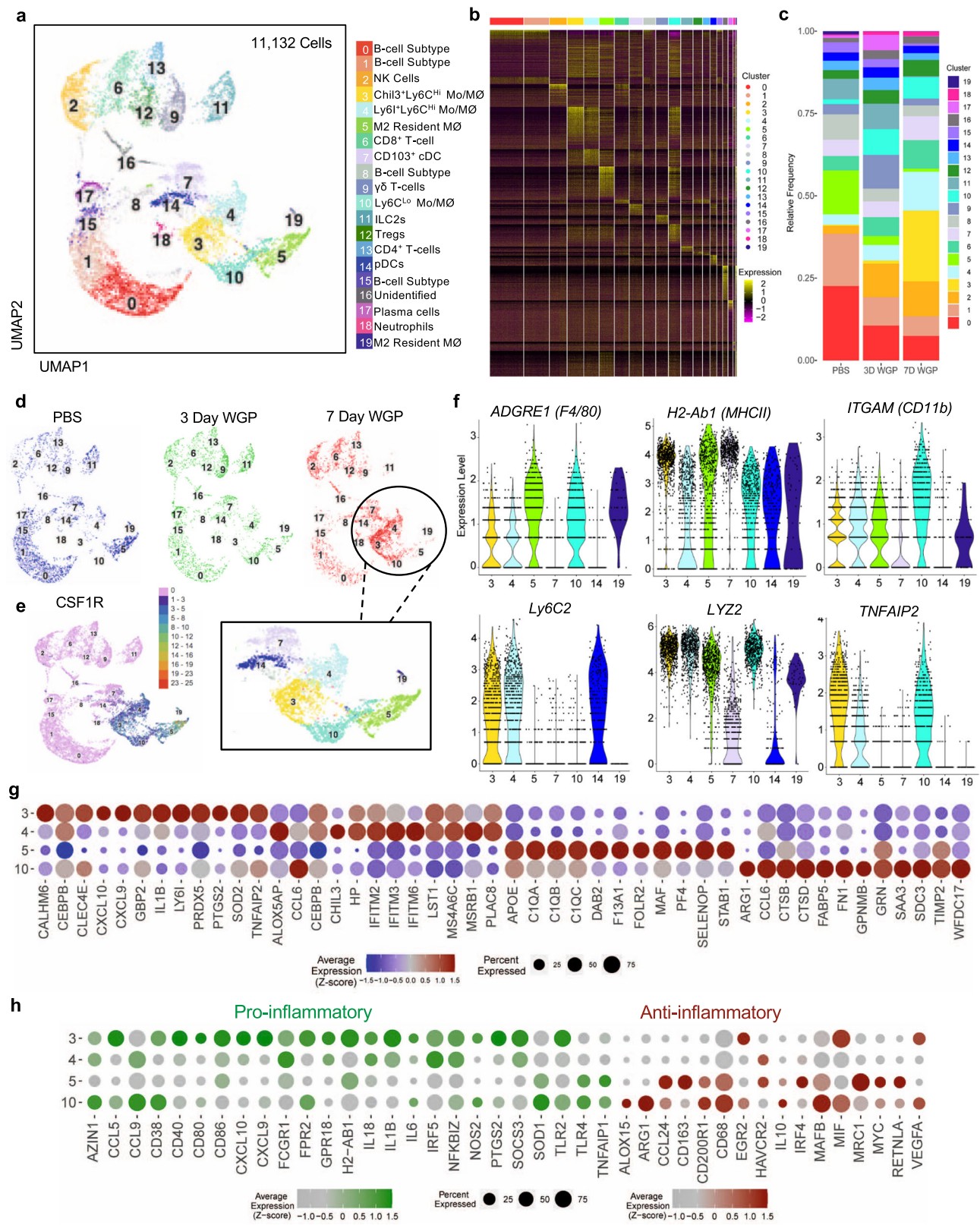

proliferating cells were CD11b⁺CCR2⁺ (Fig. 4g). We then investigated the contribution of CCR2⁺ cells to the trained phenotype. 7 days after in vivo treatment with PBS or WGP, CCR2⁺, and CCR2⁻ populations were measured for a trained response (Fig. 4h). This data indicated that the majority of cells trained following WGP treatment were CCR2⁺. To further examine the role of CCR2 in β-glucan trained monocytes/ macrophages, CCR2⁻/⁻ mice were trained with WGP β-glucan. CCR2⁻/⁻ mice did not undergo an influx of CD45⁺ (Fig. 4i) or CD45⁺CD11b⁺ myeloid cells (Fig. 4j) into the pancreas, and did not show a trained response as revealed by TNF-α production (Fig. 4k). This data indicates that CCR2 plays a critical role in the migration of innate immune cells to the pancreas and in the induction of peripheral trained immunity in the pancreas.

**Fig. 3 Single-cell RNA-seq showing the immune cell phenotype 3 and 7 days following IP WGP.** CD45[+] cells from the pancreas of mice treated with PBS or WGP 7 days and 3 days prior were sorted and scRNA-Seq was performed. **a** Two-dimensional UMAP representation of 11,132 cells aggregated from the three experimental samples with 20 unique clusters resulting from k-nearest neighbors and Louvain algorithms. **b** Heatmap of expression of aggregated marker genes for all clusters. **c** Bar graphs showing the relative frequency of cells in each cluster across samples. **d** UMAP dimension reduction of PBS (blue), 3-day WGP (green), and 7-day WGP (red) samples shown individually. The portion of the UMAP representing myeloid cells is highlighted. **e** Single-cell gene expression of *CSF1R*. **f** Single-cell expression distributions across clusters identified as myeloid cells for select genes related to myeloid phenotyping. **g** Dot plot of the top 12 enriched genes in clusters 3, 4, 5, and 10 showing the average expression level and percentage of cells expressing select genes. **h** Dot plots showing the enrichment of selected genes associated with pro-inflammatory (green) and anti-inflammatory (red) immune responses across clusters 3, 4, 5, and 10. In **g** and **h**, the average expression level is displayed as *z*-scores computed across the four clusters for individual genes.

As previously noted, CCR2 is important for the early recruitment of monocytes, but less so for late recruitment[33]. Thus far, we had also observed that at 7 days following WGP training, CCR2 was critical to the recruitment of trained myeloid cells to the pancreas. We next examined whether this influx occurred at an early time point upon WGP training and whether the CCR2[+] myeloid cells were recruited to the pancreas. To this end, mice were treated with PBS or WGP and pancreatic tissues were analyzed 24 and 48 h later. Surprisingly, as early as 24 h after WGP training, increases in CD45[+] (Supplementary Fig. 4a), CD11b[+] (Supplementary Fig. 4b), F4/80[+] (Supplementary Fig. 4c), Ly6C[+] (Supplementary Fig. 4d) and CD11b[+]CCR2[+] (Supplementary Fig. 4e) cells were observed. Additionally, it was shown that these myeloid cells were trained as early as 24 h following WGP treatment (Supplementary Fig. 4f). This observation highlights a divergence between these results and previously observed phenotypes of trained immunity which usually are not initiated until at least 3 days following training.

**WGP-trained pancreatic infiltrating myeloid cells exhibit trained responses to factors secreted from pancreatic cancers and enhanced phagocytosis and ROS-mediated cytotoxicity to tumor cells.** As we have shown that using WGP as an initial stimulus results in innate immune cells that are trained to respond more robustly to secondary exposure of LPS and that these cells accumulate in the pancreas following treatment, we wanted to know whether secondary stimuli related to pancreatic tumors may also elicit this trained response. To this end, we first investigated whether pancreatic cancer cells themselves are capable of eliciting the WGP-induced trained response. To probe this question, peritoneal macrophages were cultured with PBS or WGP in vitro and 7 days later were restimulated with LPS, the supernatant from cells cultured from a naïve mouse pancreas, and the supernatant from cultured KPC cells, which are a cell line of a pancreatic tumor on a C57BL/6 background derived from the *LSL-Kras[G12D/+]*; *LSL-Trp53[R172H/+]*; *Pdx1-Cre* (KPC) mice or Pan02 pancreatic cancer cells for 24 h. TNF-α production in the supernatant was measured by ELISA. Compared to the supernatant from cultured normal pancreatic cells which did not activate previously trained macrophages, the supernatant from cultured KPC and Pan02 pancreatic cancer cells stimulated more TNF-α production in β-glucan trained macrophages (Supplementary Fig. 5a). Additionally, to control for the possibility that the process of phagocytosis itself may cause cells to become activated, peritoneal macrophages were cultured with 3 μm polystyrene microparticle beads and were then restimulated with PBS or LPS (Supplementary Fig. 5b). Results showed that enhanced production of TNFα was specific to WGP training.

This was then tested in an ex vivo setting in which mice were treated with PBS or WGP and 7 days later the pancreases were harvested, pancreatic myeloid cells were enriched, and then restimulated with the supernatants from cultured KPC (Supplementary Fig. 5c) or Pan02 cells (Supplementary Fig. 5d). It

showed that WGP in vivo trained CD11b[+] myeloid cells in the pancreas produced significantly more TNF-α in response to tumor-conditioned media. Tumor cells themselves secrete a multitude of factors that may specifically function as the second stimulus in trained immunity. It is known that pancreatic tumor cells express high levels of damage-associated molecular patterns (DAMPs) and pro-inflammatory factors, such as macrophage migration inhibitory factor (MIF). MIF is a cytokine that is known to be secreted in high concentrations by pancreatic tumors that can act directly on myeloid cells[34,35]. Indeed, MIF was present in the supernatant of KPC and Pan02 cells as assessed by ELISA (Supplementary Fig. 5e). We thus hypothesized that MIF might be a potential tumor-secreted factor that has the capacity to act as a second signal in the setting of WGP-induced trained immunity. To investigate this, pancreatic myeloid cells from in vivo WGP-trained mice were restimulated with a similar concentration of recombinant MIF (rMIF) as present in tumor-conditioned media. As shown in Supplementary Fig. 5f, pancreatic myeloid cells previously trained with WGP showed enhanced TNF-α production upon rMIF restimulation. Collectively, these data suggest the concept that pancreatic tumor cells, through soluble factors that they release, can serve as the second signal to activate myeloid cells in the pancreas that have been trained by WGP.

We next examined whether these WGP-trained innate immune cells have enhanced intrinsic anti-tumor properties. RNA Sequencing data indicated that phagocytosis-related mechanisms were upregulated in the WGP setting (Fig. 5a), which we hypothesized could be one mechanism of anti-tumor functionality. The top 20 enriched KEGG pathways and Gene Ontology (GO) biological processes upon WGP training are listed in Supplementary Tables 1 and 2. CD45[+] pan immune cells and CD11b[+] myeloid cells from WGP-trained mouse pancreas were harvested and assayed for phagocytotic potential. WGP treatment led to a significant increase in the phagocytic potential of overall CD45[+] immune cells (Fig. 5b) and in CD11b[+] myeloid cells (Fig. 5c). In addition, in vivo trained myeloid cells showed an increase in the phagocytosis of KPC tumor cells (Fig. 5d). We then assessed whether myeloid cells trained by WGP show increased cytotoxicity to KPC cells. RNA-Seq data indicated that DEGs related to reactive oxygen species (ROS) biosynthetic processes and positive regulation of ROS metabolic processes were significantly enriched in WGP-treated myeloid cells (Fig. 5e). As a result, we hypothesized that the upregulation of ROS production by WGP would result in increased pancreatic myeloid cell cytotoxicity to KPC tumor cells. To explore this hypothesis, CD11b[+] cells from the pancreas were isolated from β-glucan trained or untrained mice and were plated with luciferase-expressing KPC cells (KPC[Luc+]) for 24 h. WGP-trained myeloid cells showed a threefold increased cytotoxicity towards KPC tumor cells, and the inhibition of ROS production using the ROS inhibitor *N*-Acetyl Cysteine (NAC) completely abrogated the WGP-elicited increase in cytotoxicity (Fig. 5f). ROS-induced lipid

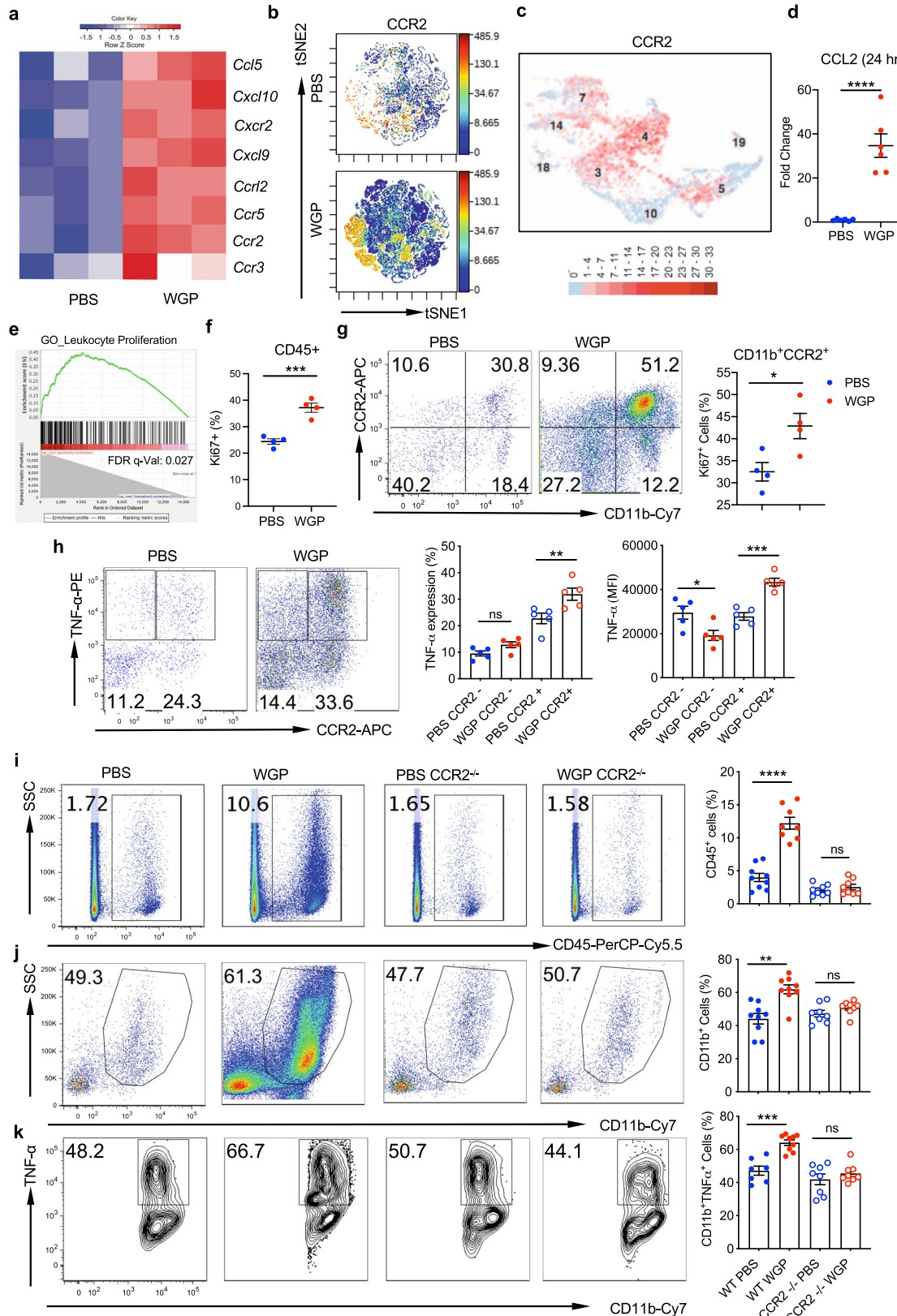

peroxidation plays a critical role in apoptosis, autophagy, and ferroptosis[36]. We thus used two additional ROS inhibitors, Trolox and deferoxamine (DFO), to examine their effect on cytotoxicity. The addition of Trolox and DFO significantly inhibited WGP-trained CD11b[+] cell-mediated cytotoxicity (Fig. 5g), suggesting

that ROS-induced lipid peroxidase is involved, at least partly, in this ROS-mediated cytotoxicity. Ultimately, this data identifies that pancreatic tumor cells are capable of reactivating WGP-trained infiltrating myeloid cells in the pancreas and that these cells show enhanced phagocytosis and ROS-mediated cytotoxicity.

**Fig. 4 CCR2 is required for immune cell trafficking into the pancreas. a** Heatmap of chemokines and cytokines that were upregulated in 7-day WGP-treated CD11b$^+$ cells based on RNA-Seq data. **b** viSNE plot of the CD11b$^+$ pancreatic population in PBS and 7-day WGP-trained mice, highlighting the expression of CCR2. Images made with CyTOF data. **c** scRNA-Seq data showing a UMAP of the myeloid clusters expressing CCR2. **d** CCL2 expression in whole pancreatic lysates 24 h following WGP treatment a measured by RT-PCR ($n = 6$). ****$p = 0.0001$. **e** GSEA generated enrichment plots of genes related to leukocyte proliferation in CD11b$^+$ pancreatic cells from 7-day WGP-trained as compared to PBS mice. PBS was used as the control and compared to WGP. **f** Summarized data of the percent of CD45$^+$ pancreatic cells that are Ki67+ in PBS and 7-day WGP-trained mice ($n = 4$). ***$p = 0.0007$. **g** Cells were first gated on the CD45 + Ki67+ population. Plots show the percent of the CD45$^+$Ki67$^+$ proliferating pancreatic cells that are CD11b$^+$CCR2$^+$ in PBS and 7-day WGP-trained mice ($n = 4$). *$p = 0.0261$. **h** Pancreatic cells from PBS and 7-day WGP-trained mice ($n = 5$) were restimulated with LPS and the percent of CCR2 positive and negative cells producing TNFα was measured in each condition. Cells were first gated on the CD45$^+$CD11b$^+$ subset. Representative dot plots and summarized data were shown. *$p = 0.022$, **$p = 0.0073$, ***$p = 0.0007$. **i–k** WT and CCR2$^{-/-}$ mice were treated with PBS or WGP and the percent of **i** CD45$^+$ cells in the pancreas (****$p < 0.0001$) and **j** the percent of CD45$^+$ cells that were CD11b$^+$ was assessed. **$p = 0.0013$. **k** The percent of CD45$^+$CD11b$^+$ cells producing TNFα. Representative flow plots and summarized data are shown. Each dot represents data from one mouse. ***$p = 0.0002$. (**i–k**: WT PBS $n = 9$, WT WGP $n = 9$, CCR2$^{-/-}$ PBS $n = 8$, CCR2$^{-/-}$ WGP $n = 8$). An unpaired, two-tailed student's $t$-test was used in **d**, **f**, and **g**, while a one-way ANOVA with multiple comparisons was used in **h–k**. Data were represented as mean ± SEM. ns not significant. In **f–k**, each sample represents a biologically independent animal obtained over a single independent experiment which was repeated at least twice for verification of results.

**WGP-induced trained immunity reduces tumor growth and prolongs survival in orthotopic models of pancreatic cancer.** We next investigated whether β-glucan-mediated trained innate immune responses in the pancreas would result in the establishment of an anti-tumor microenvironment that may be sufficient to overcome the characteristically immunosuppressive TME of PDAC. Accordingly, mice were given 1 dose of either PBS or WGP on day −7, and on day 0, $1 \times 10^5$ KPC or KPC$^{Luc+}$ cells were orthotopically implanted into the tail of the pancreas (Fig. 6a). On day 21 mice were euthanized and the tumor weight was measured (Fig. 6b). In the setting of injection of KPC$^{Luc+}$ cells, mice were injected with luciferase substrate, and tumors were imaged (Fig. 6c). Both studies showed a remarkable reduction in tumor burden as a result of WGP treatment, and survival was also significantly prolonged in mice trained with WGP (Fig. 6d). Immunophenotyping of the tumors showed a persistent increase in CD45$^+$ immune cells, CD11b$^+$ myeloid cells, and F4/80$^+$ macrophages in the WGP-trained setting (Fig. 6e). CD11b$^+$ myeloid cells (Fig. 6f) and CD11b$^+$F4/80$^+$ macrophages (Fig. 6g) within the tumor also showed a significant increase in TNF-α production due to WGP. Both an increased number of CD11b$^+$ myeloid cells (Fig. 6h, left) and an increase in the percent of CD11b$^+$ cells producing TNF-α (Fig. 6h, right) significantly correlated with decreased tumor burden. Neither CD4$^+$ nor CD8$^+$ T-cells showed increased IFN-γ, further supporting that the reduction in tumor burden was driven by the WGP-trained myeloid cells (Fig. 6i). Similar anti-tumor effects were also observed in mice orthotopically implanted with Pan02 tumors (Supplementary Fig. 6a). To further confirm that innate immune cells are responsible for the observed anti-tumor immune responses, orthotopic KPC tumors were implanted into NSG mice. Similar to WT mice, NSG mice also showed a significant reduction in tumor size due to WGP training, confirming that the anti-tumor effects of WGP were driven by innate immune cells and functioned independently of adaptive responses (Supplementary Fig. 6b). Kalafati et al. had shown that innate immune training of granulopoiesis promotes anti-tumor immunity[21]. While we had not seen an important contribution of granulocytes to our phenotype of trained immunity, we examined whether granulocytes are involved in WGP-dependent reduction in pancreatic tumors by depleting neutrophils and observing tumor growth in PBS and WGP-treated mice. The depletion efficiency of neutrophils in the pancreas (Supplementary Fig. 6c) along with the pancreatic tumor burden were assessed (Supplementary Fig. 6d). Our results showed a significant reduction in tumor size in the WGP group in the absence of neutrophils.

**Trained CCR2$^+$ myeloid cells are a primary effector cell in the anti-tumor mechanism.** Given the complete inhibition of innate immune cell trafficking into the pancreas and training of pancreatic myeloid cells following WGP treatment in CCR2$^{-/-}$ mice, we reasoned that CCR2$^{-/-}$ mice would also not show the beneficial anti-tumor immune effects of WGP training. In line with this hypothesis, CCR2$^{-/-}$ mice that received WGP did not show a reduced tumor burden (Fig. 7a) as compared to WT mice. This demonstrated that CCR2 is requisite for the WGP-driven influx of trained innate immune cells into the pancreas and that those are consequential for the anti-tumor effects. Though CCL2-CCR2 signaling had been identified to be critical in the recruitment of trained monocyte-derived macrophages to the pancreas, we also wanted to know whether the presence of trained HSCs in the bone marrow and the generation of centrally trained immunity alone was sufficient to slow the growth of orthotopic pancreatic tumors.

To further confirm the direct anti-tumor functionality of the infiltrating CCR2$^+$ myeloid cells, the CCR2$^+$ and CCR2$^-$ myeloid populations from WGP-trained mice were sorted, admixed with KPC tumor cells, and implanted orthotopically into mice. Tumors admixed with CCR2$^+$ cells were smaller than those that were admixed with CCR2$^-$ cells, further supporting that the trained CCR2$^+$ myeloid cells themselves are a primary effector cell in the anti-tumor mechanism (Fig. 7b). CyTOF analysis of these tumors revealed that the CCR2$^+$ admixed tumors had significantly fewer CD11b$^+$ myeloid cells and significantly increased CD8$^+$ T-cells present within the tumor (Fig. 7c, d). The ratio of CD8$^+$ T-cells: CD11b$^+$ myeloid cells was also significantly increased in the CCR2$^+$ admix condition (Fig. 7e).

**WGP synergizes with anti-PD-L1 mAb therapy to prolong survival in models of PDAC.** Although a significant reduction in tumor burden was shown due to WGP training, it is the case that all mice developed fatal tumors. Our studies had identified that WGP treatment drastically impacted the phenotype of the myeloid populations in the pancreas. Though we had identified that the primary effector cell responsible for the anti-tumor effects of WGP are CCR2$^+$ infiltrating monocytes/macrophages, we reasoned that these immune changes may also impact the overall TME in a way that could make adaptive immune cells more responsive to checkpoint blockade therapy. Specifically, because we had observed an increase in the proportion of CD8$^+$ T-cells present within CCR2$^+$ admixed tumors (Fig. 7c) and had also observed significant PD-L1 expression on myeloid cells present within KPC tumors (Fig. 7f), we hypothesized that WGP treatment may potentiate the effects of anti-PD-L1 mAb therapy.

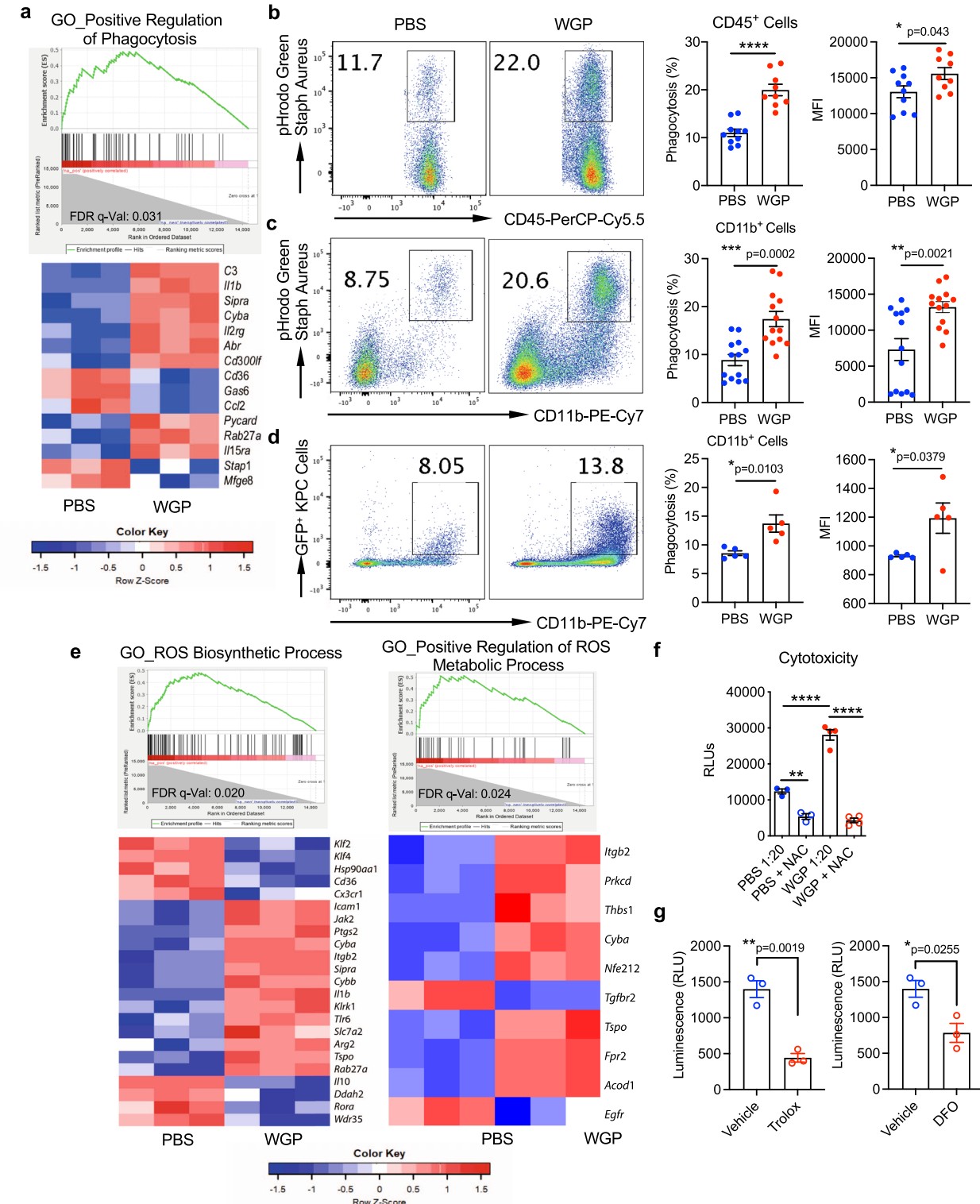

To assess whether anti-PD-L1 mAb therapy synergizes with WGP-induced trained immunity in the pancreas, PBS or WGP-treated mice that were implanted with orthotopic KCP tumors were then given either anti-PD-L1 mAb or rat IgG2b isotype control mAb. As has been shown in several clinical trials, anti-PD-L1 therapy alone failed to prolong survival even beyond that of the IgG2b isotype control mAb treated mice (Fig. 7g). WGP-trained mice survived significantly longer than IgG2b isotype and anti-PD-L1treated mice. However, a combination of WGP and anti-PD-L1 together prolonged survival most effectively. This shows that there is a clinical benefit to combining WGP with anti-PD-L1 immuno-checkpoint blockade therapy.

Thus far, the use of WGP has been described in a setting in which WGP is administered before tumor cells are implanted. Considering the reduction in tumor size of this model, we also tested a more clinically relevant model in which mice were implanted with orthotopic KPC tumors and WGP was used to incite trained immunity thereafter Fig. 7h). In the therapeutic setting, the WGP-driven influx of trained myeloid cells to the pancreas was also shown to prolong survival. Together, these data

**Fig. 5 WGP-trained pancreatic infiltrating myeloid cells show enhanced phagocytosis and ROS-mediated cytotoxicity. a** Enrichment plots (GSEA) and heatmap of genes related to the positive regulation of phagocytosis in CD11b$^+$ cells from 7-day WGP-trained as compared to PBS mice. **b** The percent of CD45$^+$ pancreatic cells that phagocytosed a pHrodo Green Staph Aureus particle in PBS ($n = 10$) and 7-day WGP ($n = 9$) mice along with the MFI of the pHrodo Green Staph Aureus particle. Each dot represents data from one mouse. *$p = 0.043$, ****$p < 0.0001$. **c** The percent of CD11b + myeloid pancreatic cells that phagocytosed a pHrodo Green Staph Aureus particle in PBS ($n = 13$) and 7-day WGP ($n = 13$) mice. Cells were first gated on the CD45$^+$ population. The MFI of the pHrodo Green Staph Aureus particle is shown. **$p = 0.0021$, ***$p = 0.0002$. **d** The percent of CD11b$^+$ myeloid pancreatic cells that phagocytosed KPC$^{GFP+}$ tumor cells in PBS ($n = 5$) and 7-day WGP ($n = 5$) mice. *$p = 0.0103$. Cells were first gated on the CD45$^+$ population. The summarized MFI of the GFP$^+$ signal is also shown. *$p = 0.0379$. **e** Enrichment plots (GSEA) and heatmap of genes related to the reactive oxygen species biosynthetic processes and the positive regulation of reactive oxygen species metabolic processes in CD11b$^+$ cells from 7-day WGP-trained as compared to PBS mice. **f** Summarized results from a cytotoxicity assay where CD11b$^+$ cells from PBS and 7-day WGP-trained mice were sorted and incubated at a ratio of 1:20 KPC$^{Luc+}$: CD11b$^+$ cells for 24 h. NAC was used to block ROS expression and the tumor cytotoxicity was assessed by luminescence (PBS $n = 3$, WGP $n = 4$). **g** Summarized data from cytotoxicity assays where CD11b$^+$ cells from 7-day WGP-trained mice were sorted and incubated at a ratio of 1:20 KPC$^{Luc+}$: CD11b$^+$ cells for 24 h. Trolox or DFO or respective vehicle controls were added and tumor cytotoxicity was determined by measuring luminescence ($n = 3$). *$p = 0.0255$, **$p = 0.0019$. An unpaired, two-tailed student's $t$-test was used in **b**–**d** and **g** and a one-way ANOVA with multiple comparisons was used for **f**. Data were represented as mean ± SEM. ns not significant. In **b**–**g**, each sample represents a biologically independent animal obtained over a single independent experiment which was repeated at least twice for verification of results.

suggest that the initiation of trained immunity in the pancreas using WGP has relevant clinical applications in treating pancreatic cancer that necessitate further translational research and investigation.

## Discussion

While the trafficking of β-glucan has been previously characterized, the specific tropism of β-glucan to the pancreas has not been previously reported[37]. We show that particulate β-glucan can traffic directly into the pancreas and can also be phagocytosed by macrophages which then traffic into the pancreas. The relationship between the peritoneal cavity and the pancreas has not been well defined, and studies relating to the pathophysiology of acute pancreatitis (AP) have identified that peritoneal macrophages are a principal contributor to the inflammatory response in AP, thus supporting a connection between the peritoneum and the pancreas[38]. Our imaging data indeed show that peritoneal macrophages that phagocytose isotope-labeled WGP primarily traffic to the pancreas. Studies on liver injury have further supported this model of a dynamic interchange of cells between the peritoneal cavity and solid organs that is independent of the circulation[39]. Together this suggests that even in homeostatic conditions there exists a basal level of immune cell exchange between the pancreas and peritoneal cavity that can be exploited in the setting of particulate β-glucan. While this study investigated particulate β-glucan WGP, it may also be interesting to investigate whether soluble yeast-derived β-glucan shows a similar tropism and signaling mechanism within the pancreas. Particulate β-glucans are known to signal directly through Dectin-1, while soluble β-glucans are thought to function through a CR3-dependent pathway to exert their anti-tumor properties[40,41]. One study showed that IP administration of soluble β-glucan resulted in trafficking to the peripheral blood and APCs in the peritoneum, spleen, and bone marrow, indicating that despite the different molecular signaling mechanisms, a similar ability to activate trained immunity may be possible[42]. However, given that our data indicates that this trafficking is Dectin-1 dependent, it is possible that only particulate β-glucan will display direct trafficking to the pancreas though both particulate and soluble may initiate mechanisms of trained immunity. Further, previous studies which characterize the trafficking of intravenously injected soluble β-glucan show trafficking to the spleen, kidney, and liver though do not mention tropism to the pancreas[43]. As the pancreas has not been previously thought to be a target of β-glucan trafficking it is not clear whether the pancreas simply has never been studied or whether pancreatic trafficking does not occur.

Future studies should address the biodistribution of both soluble and particulate β-glucan across various routes of administration.

β-Glucan trafficking to the pancreas has a multifactorial impact on the immune populations present within the pancreas. First, β-glucan arrival to the pancreas directly impacts the populations of immunosuppressive M2 resident macrophages present within the pancreas that are known to be important in the promotion of pancreatic tumors. CyTOF and scRNA-Seq data showed nearly complete disappearance of the resident macrophage population 7 days following WGP administration. Interestingly, the disappearance of the resident macrophage population coincides directly with a reciprocal appearance of a Ly6C$^{lo}$ macrophage population. This Ly6C$^{lo}$ macrophage population bears similar phenotypic markers to the resident population, though skews more towards an M1 phenotype. It is thus likely that resident macrophages come into contact with β-glucan that has trafficked to the pancreas and these cells become repolarized, therefore taking on a different cellular phenotype which results in the formation of a unique cluster. Second, the arrival of β-glucan to the pancreas results in amplified chemokine/chemokine receptor signaling which recruits pro-inflammatory Ly6C$^{Hi}$ infiltrating monocyte-derived macrophages from the periphery to the pancreas. Twenty-four hours following the administration of WGP, CCL2 levels in whole pancreatic lysates are found to be increased by 30-fold. Accordingly, we show that the robust β-glucan-dependent cellular influx to the pancreas is dependent on CCR2.

While we focus on the dynamics of the myeloid compartment due to the observed importance of CCR2/CCL2 signaling, there was an interesting initial enhancement of cluster 11 on day 3 and then disappearance by day 7 following WGP treatment. Cluster 11 was characterized as ILC2s based on the expression of several markers such as *KLRG1*, *IL-5*, *IL-13*, and *ICOS*. ILC2s are known to be long-term tissue-resident cells[44]. Recent studies have shown that activation of local ILC2s by tissue-specific alarmins induces their proliferation, lymph node migration, and blood dissemination. ILC2s from several tissues, such as the gut and the lung, have been shown to enter the blood by extrusion from these perturbed tissues and migrate to other tissues such as the liver[44–46]. Such a mechanism could explain the initial increase and then disappearance of ILC2s in the pancreas. Given that ILC2s have recently been implicated in the response of pancreatic tumors to immunotherapy[47], more research is warranted that investigates the impact of WGP and the initiation of trained immunity on the ILC2 population of the pancreas.

Although the initiation of an influx of pro-inflammatory immune cells to the pancreas is in itself important, it is that the CCR2$^+$Ly6C$^{Hi}$ infiltrating monocyte-derived macrophage

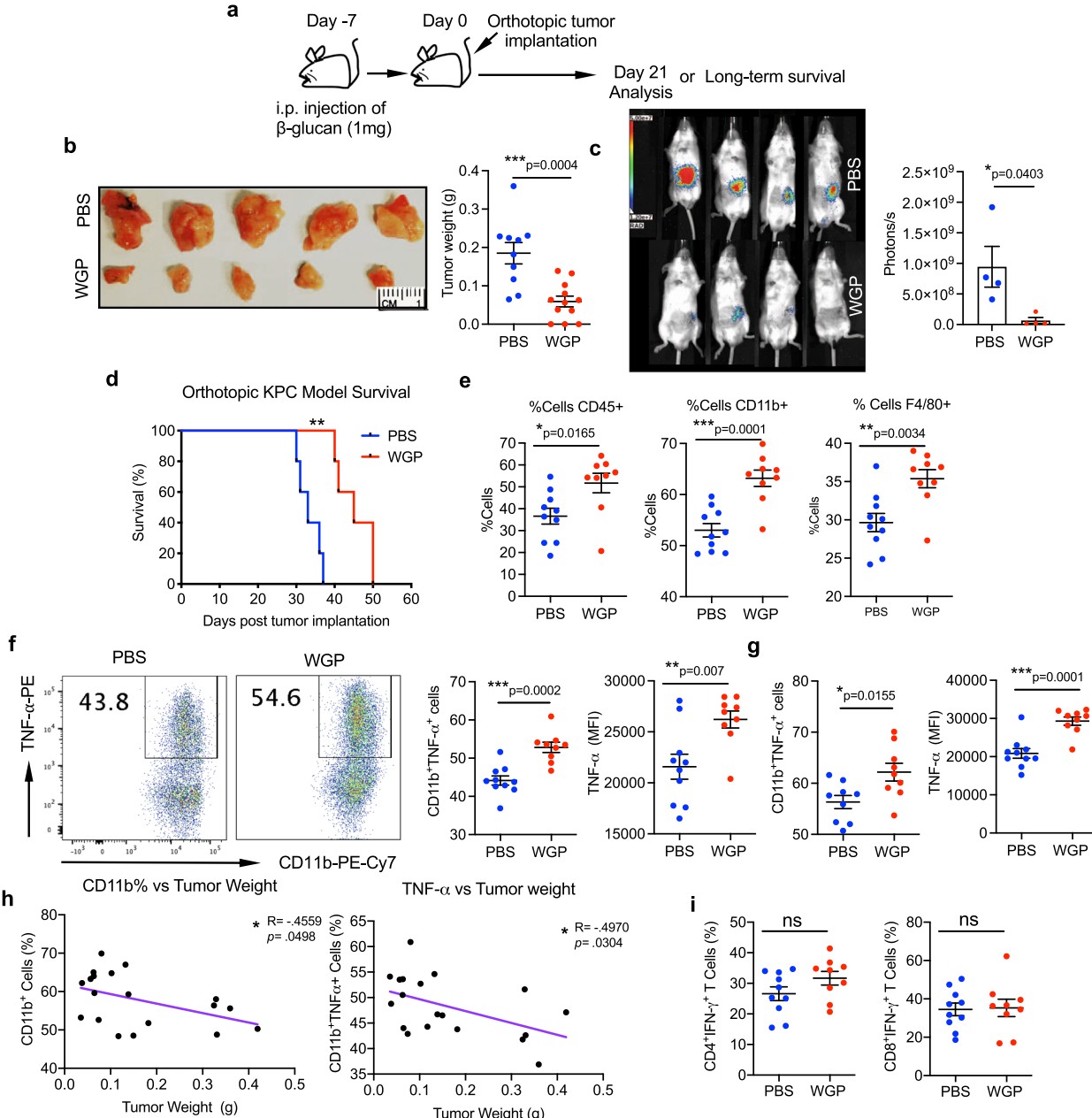

**Fig. 6 The induction of trained immunity in the pancreas has anti-tumor effects. a** Experimental schema. **b** C57BL/6 mice received a single IP injection of WGP or PBS and 7 days mice were implanted orthotopically with KPC pancreatic cancer cells. Representative pictures of tumors and quantitative analysis of tumor weight are shown. Tumor weight was measured at day 21 (PBS $n = 10$, WGP $n = 12$). ***$p = 0.0004$. **c** C57BL/6 mice ($n = 4$) received a single i.p. injection of WGP or PBS and 7 days later were implanted orthotopically with KPC$^{+Luc}$ pancreatic cancer cells. On day 21 post tumor implantation, mice were given I.P. luciferin bioluminescent substrate and were placed in a photon imager to measure tumor size in vivo. *$p = 0.0403$. **d** Survival of mice in the experimental schema shown in **a**, using KPC cells ($n = 5$). **$p = 0.0018$. **e** Phenotyping of the tumors showing the percent of viable cells that are CD45$^+$, the percent of the CD45$^+$ population that are CD11b$^+$, and the percent of CD11b$^+$ cells that are F4/80$^+$ (PBS $n = 10$, WGP $n = 9$). *$p = 0.0165$ (CD45), ***$p = 0.0001$ (CD11b), **$p = 0.0034$ (F4/80). **f** TNFα production in CD11b$^+$ cells from PBS and 7-day WGP-trained that were restimulated with LPS. Percent of TNFα$^+$ cells and the MFI of TNFα are shown (PBS $n = 10$, WGP $n = 9$). **$p = 0.007$, ***$p = 0.0002$. **g** Summarized data of TNFα production in CD11b$^+$F4/80$^+$ cells from PBS and 7-day WGP-trained that were restimulated with LPS. Percent of TNFα$^+$ cells and the MFI of TNFα are shown (PBS $n = 10$, WGP $n = 9$). *$p = 0.0155$, ***$p = 0.0001$. **h** Tumor weight was correlated with the percent of CD45$^+$ immune cells that were CD11b$^+$ (left) and the percent of CD45$^+$CD11b$^+$TNFα$^+$ cells (right). (PBS $n = 10$, WGP $n = 9$). **i** Summarized data of the percent of CD4 + and CD8 + T-cells expressing IFNγ (PBS $n = 10$, WGP $n = 9$) Data were represented as mean ± SEM. Pearson correlation coefficients were used to measure the strength of the linear associations and unpaired, two-tailed student's $t$-tests were used otherwise. ns not significant. Each sample represents a biologically independent animal obtained over a single independent experiment which was repeated at least twice for verification of results.

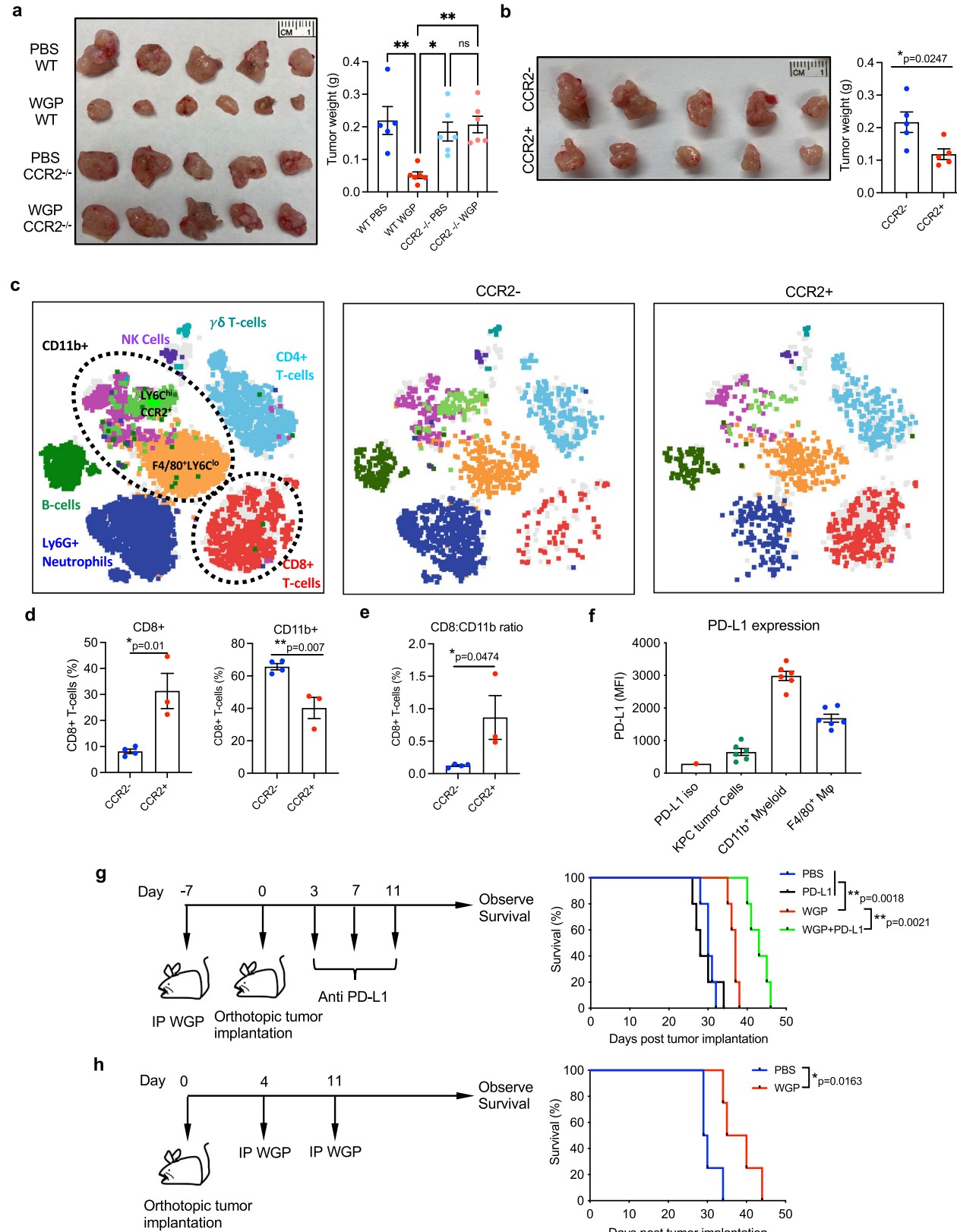

populations from the periphery display features of trained immunity which carries the most important implications as the induction of peripheral trained immunity in the pancreas has not yet been characterized. This induced trained immunity is also associated with metabolic and epigenetic reprogramming. While CCR2 is known to be an important receptor in the recruitment of monocytes, this is also the first indication that CCR2 signaling on monocytes is requisite for the establishment of peripheral trained immunity. It is noted that CCR2[+] monocytes/macrophages have been linked to tumor metastasis and progression[48,49]. However, our data suggest that the recruitment of reprogrammed, trained CCR2[+] monocytes/macrophages exert robust anti-tumor effects.

**Fig. 7 The anti-tumor effector mechanisms of WGP treatment and clinically relevant models. a** WT and CCR2$^{-/-}$ mice were treated with PBS or WGP and 7 days later were implanted with orthotopic KPC pancreatic tumor cells. Tumor weight at day 21 is reported. (WT PBS $n = 5$, WT WGP $n = 6$, CCR2 PBS $n = 6$, CCR2$^{-/-}$ WGP $n = 6$). **$p = 0.0027$ (WT PBS vs WT WGP), *$p = 0.0118$ (WT WGP vs CCR2$^{-/-}$ PBS), **$p = 0.0034$ (WT WGP vs CCR2$^{-/-}$ WGP). **b** Sorted CCR2$^{+}$ and CCR2$^{-}$ pancreatic CD11b$^{+}$ cells from WGP-trained mice were admixed with KPC cells and implanted orthotopically. Tumor size was evaluated 21 days later (CCR2$^{+}$ $n = 5$, CCR2$^{-}$ $n = 5$). *$p = 0.0247$. **c** t-SNE plots generated by CyTOF analysis of the admix tumors from **b**. Clusters showing significant differences between groups are indicated by the circles. Total data (left) and representative t-SNE plots of each group are shown. **d** Summarized percent of CD8$^{+}$ and CD11b$^{+}$ cells in admix tumors (PBS $n = 4$, WGP $n = 3$). *$p = 0.01$, **$p = 0.007$. **e** The ratio of CD8$^{+}$:CD11b$^{+}$ cells in admix tumors (PBS $n = 4$, WGP $n = 3$). *$p = 0.0474$. **f** Expression of PD-L1 on KPC tumor cells, CD11b$^{+}$ and F4/80$^{+}$ cells in a KPC tumor 21 days after implantation (KPC $n = 5$, CD11b$^{+}$ $n = 6$, F4/80$^{+}$ $n = 6$). **g** Experimental schema of WGP and anti-PD-L1 therapy. Mice ($n = 5$) were treated with PBS or WGP and 7 days later were implanted with orthotopic KPC pancreatic tumors. On days 3, 7, and 11 post-implantation, mice were given anti-PD-L1 mAb or anti-rat IgG2b mAb isotype control. Survival was monitored. **$p = 0.0018$ (WGP vs PBS or PD-L1), **$p = 0.0021$ (WGP vs WGP + PD-L1). **h** Experimental schema of WGP used in the therapeutic setting. Mice were implanted with orthotopic KPC pancreatic tumors and were given WGP once mice had recovered from the surgery at day 4, and 1 week later on day 11. *$p = 0.0163$. Data were represented as mean ± SEM. A one-way ANOVA with multiple comparisons was used for **a** and an unpaired, two-tailed student's $t$-test was used for **b**, **d**, and **e**. Log-rank test was used for **g** and **h**. ns not significant. Each sample represents a biologically independent animal obtained over a single independent experiment which was repeated at least twice for verification of results.

This is directly demonstrated by our admix experiment where mice that received CCR2$^{+}$ myeloid cells from WGP-trained mice showed a significantly reduced tumor burden as compared to mice that received CCR2$^{-}$ myeloid cells from the same trained mouse.

As we show that peripheral trained immunity has been established in the pancreas, we ask whether these cells could be reactivated by tumor cells or their secreted factors. Re-exposure of in vivo WGP-trained pancreatic myeloid cells to tumor-conditioned media was shown to vigorously elicit a trained response. Kalafati et al. recently showed a role for granulocytes in trained immunity driven anti-tumor mechanisms[21], however, given their use of subcutaneous models of cancer, the currently reported study is the first instance suggesting that myeloid cells in a specific organ can be trained to react directly to tumor cells of that same organ. It has been well established that myeloid cells are capable of directly killing tumor cells through mechanisms of phagocytosis and ROS production[50–54], and here we show that WGP training significantly upregulates the direct anti-tumor functionalities of enhanced phagocytosis of tumor cells and ROS-mediated cytotoxicity to tumor cells.

Further, while we highlight that tumor-conditioned media can reactivate trained myeloid cells, we also identify a specific factor, MIF, in the tumor-conditioned media that is involved in this activation. These data suggest that the induction of trained immunity could participate in mechanisms of tumor immunosurveillance. For example, in the early stages of tumor development, tumors secrete soluble factors that can re-stimulate trained innate cells leading to the eradication of these tumor cells.

We then translate this understanding of the anti-tumor potential of the trained innate myeloid cells that enter the pancreas to orthotopic models of PDAC, where we observe a dramatic decrease in the tumor burden and an increase in the survival of mice given only one administration of WGP. We further demonstrate that the cells responsible for tumor control are mainly trained CCR2$^{+}$ myeloid-derived cells. In contrast to other proposed models of tumor control which appear to be entirely driven by the induction of central trained immunity in the BM[21], instead, in this model the early trafficking of WGP to the pancreas and the resulting production of CCL2 in the pancreas recruits trained CCR2$^{+}$ myeloid cells from the periphery which are mainly responsible for the anti-tumor effects. WGP directly trafficking to the pancreas also repolarizes M2-like tissue-resident macrophages towards M1-like macrophages that may also restrain tumor progression. While the tumor reduction observed in mice implanted with KPC tumors that had been admixed with CCR2$^{+}$ trained cells was significant, it was not as striking as mice given IP WGP. This supports that the anti-tumor effects of IP WGP are twofold; the recruitment of CCR2$^{+}$ trained anti-tumor myeloid cells to the pancreas and the repolarization of pro-tumorigenic resident macrophages work in concert to elicit anti-tumor innate responses that lead to reduced tumor burden in models of PDAC.

We also note the high expression of MHCII on these trained monocyte/macrophages, which likely plays a role in tumor antigen processing and communication with T-cells to elicit adaptive immune responses against the tumor. Interestingly, while we confirmed that these anti-tumor mechanisms do not depend on adaptive immune responses, we did observe a significant increase in CD8$^{+}$ T-cells present within CCR2$^{+}$ admixed tumors. Additionally, given that myeloid cells in the tumors were observed to display high levels of PD-L1 and that the presence of trained CCR2$^{+}$ cells in the pancreas appeared to recruit CD8$^{+}$ T-cells, we reasoned that WGP may potentiate the effects of anti-PD-L1 therapy. In accordance with the literature, anti-PD-L1 mAb therapy alone had no survival benefit while the combination of WGP and anti-PD-L1 synergized to produce better survival outcomes than WGP alone[55]. This suggests that trained innate immune cells may activate adaptive immune cells in the TME, however, the immunosuppressive microenvironment likely needs to be further overcome by the concomitate effects of anti-PD-L1. Importantly, despite improved survival, all mice did eventually succumb to the tumor, so future investigations into the optimization of this treatment in terms of dosing and the number of treatments are warranted.

Immunosuppressive myeloid cells play a critical role in the creation of the immunosuppressive TME in PDAC. Here, we demonstrate a capability to engage these myeloid cells in the setting of PDAC through the induction of trained innate immunity in the pancreas. Importantly, we also show that the ability to initiate trained immunity in the pancreas potentiates the therapeutic effects of immunotherapy such as anti-PD-L1 immuno-checkpoint blockade therapy. Lending further clinical applicability to these findings, we also exhibit that the induction of trained immunity in the adjuvant setting can decrease tumor size and prolong survival. These findings emphasize the potential of using β-glucan to therapeutically target myeloid cells within the TME and highlight that the induction of peripheral trained immunity in the pancreas could play a consequential role in reprogramming the suppressive TME of PDAC which could result in extending the life of patients diagnosed with this deadly malignancy.

## Methods

**Ethical statement.** All experiments involving animals were conducted according to the ethical guidelines set by the University of Louisville Institutional Animal Care and Use Committee. Experiments were conducted according to the approved protocols 19471 and 19536.

**Mice.** Six- to eight-week-old male and female mice were used in all experiments. Wild-type (WT) C57BL/6 J mice were purchased from the Jackson Laboratory (Bar Harbor, ME, USA) or bred in the University of Louisville specific pathogen-free (SPF) animal facility. C57BL/6 Dectin-1 knockout (Dectin-1$^{-/-}$) mice were described previously[56]. CCR2 global knockout mice were purchased from Jackson Laboratory. Albino C57BL/6 mice were kindly provided by Dr. Jonathan Warawa at the University of Louisville. NOD/SCID/IL2ry$^{Null}$ (NSG) mice were purchased from the Jackson Laboratory. The maximal tumor size allowed by the IUCAC was no more than 2000 mm$^3$. Mice were monitored daily for tumor growth and euthanized when tumor burden exceeded the limited size. Mice were maintained on a 12-h dark/light cycle at room temperature with controlled humidity (around 55%). All animals were housed in a barrier facility and only healthy mice were used for experiments. All mice were at least 6 weeks of age upon use, and all experiments involving animals were performed in compliance with all relevant laws and institutional guidelines provided by the Rodent Rearing Facility (RRF) and approved by the Institutional Animal Care and Use Committee (IACUC) of the University of Louisville.

**Preparation of β-glucan.** Highly purified particulate β-glucan in the form of particulate whole β-glucan particles (WGP) isolated from *Saccharomyces cerevisiae* was provided by Biothera. Before use, WGP was gently sonicated for 15 s, two times using a Qsonica Q55-110 Q55 Sonicator (Cole-Parmer) to ensure aggregates were broken up.

**Preparation and use of DTAF-WGP.** DTAF (Sigma-Aldrich) at 2 mg/mL was mixed with a suspension of 20 mg/mL WGP in borate buffer (pH 10.8). This incubated at room temperature for 8 h with continuous mixing. Following incubation, the WGP was centrifuged and washed with cold sterile endotoxin-free DPBS (Sigma-Aldrich) five times or until the supernatant no longer contained visible DTAF. The concentration was adjusted to 10 mg/mL in the endotoxin-free DPBS for storage. About 1 mg of the DTAF-WGP has injected IP into C57BL/6 and Dectin-1$^{-/-}$ mice and organs were harvested 3 days later.

**Preparation of the $^{89}$Zr-WGP.** WGP (100 mg) was mixed with Deferoxamine-SCN (2.7 mg, in 0.8 ml DMSO) and suspended in 10 ml sodium carbonate buffer (0.1 M, pH 9.4) overnight at room temperature in the dark with gentle shaking. The Deferoxamine-labeled WGP was then washed with DI water (10 × 40 mL), and 30 mg of Deferoxamine-labeled WGP was mixed with 3 mCi of $^{89}$Zr oxalate in 2 ml Tris•HCl buffer (0.5 M, pH 7.5), and then incubated at 37 ºC for 60 min with shaking. The $^{89}$Zr-WGP was then centrifuged and washed with 3 ml of sterile PBS. The radioactivity of $^{89}$Zr-WGP was measured by a dose calibrator and used for in vitro and in vivo studies.

**Biodistribution and PET/CT scan using $^{89}$Zr-WGP.** Positron Emission tomography (PET)/computed tomography (CT) imaging was conducted in C57BL/6 and Dectin-1$^{-/-}$ mice 48 h after IP injection of 1 mg of pure $^{89}$Zr-WGP or injection of 1 × 10$^6$ peritoneal macrophages that had been co-cultured with 25 μg/ml of $^{89}$Zr-WGP for 2 h and then gently washed to remove excess $^{89}$Zr-WGP. The mice were scanned for 15 min with a Siemens R4 MicroPET followed by 10 min of CT scan. Siemens IAW software was used for the acquisition and reconstruction of the PET signal, and Siemens IRW software was used for merging and analyzing the imaging data. At the end of the imaging study, mice were euthanized, and organs of interest were harvested. For biodistribution, 50 uL of peripheral blood was collected using a retro-bulbar bleeding technique. The brain, heart, lungs, liver, spleen, kidneys, pancreas, large intestine, small intestine, stomach, femur, a piece of skin from the flank of the mice, and the rectus femoris muscle were harvested, weighed, and placed in a 2470 Wizard automatic gamma counter (PerkinElmer) in order to measure the radioactivity of each tissue. The CPM values were calculated using Prism software (GraphPad Software, La Jolla, CA).

**Pancreatic processing.** Following euthanization, mouse pancreas were harvested and gently cut into smaller pieces using sterile scissors. They were suspended in a 15 mL tube in complete media (RPMI) with 1X digestion buffer comprised of 300 U/ml collagenase I, 60 U/ml Hyaluronidase, and 80 U/ml DNase (Sigma). These were placed in a rotating incubator at 37 °C with 5% CO2 for 15–20 min. The digestion buffer was then quenched with ice-cold complete RPMI 1640 and washed. Cells were passed through a sterile nylon 40 μm basket filter and small undigested pieces of tissue were smashed using a sterile syringe stopper in order to generate a single-cell suspension. If an appreciable number of red blood cells (RBCs) were seen to exist in the sample, RBC lysis was performed by adding 2 mL of sterile 10x ACK (Thermo Fisher Scientific).

**In vivo WGP administration.** Mice were given a single 1 mg intraperitoneal dose of gently sonicated WGP, 3 μm polystyrene beads (Sigma-Aldrich), or 3 μm fluorescent microspheres (Polysciences) (all 1 mg in 200 μl of sterile PBS) or 200 μl of sterile PBS on day 0. For typically trained immunity studies, 7 days following the initial IP dose, mice were euthanized using CO$_2$ and the pancreas along with other tissues of interest were removed and processed. For dose-titration studies, 0.5, 1, and 2 mg of WGP were delivered IP in 200 uL of sterile PBS. For time titration studies, mice were injected with 1 mg of WGP, and the pancreas was harvested after 24 h, 48 h, 3, 7, 10, 16, and 30 days later.

**Ex vivo restimulation.** In order to assess the trained phenotype of mononuclear cells in mice treated with WGP or PBS ex vivo, after processing the pancreas, pancreatic cell suspensions were plated in 24 well plates and stimulated with LPS (10 ng/ml), the supernatant from cultured KPC and Pan02 cells (40%), and recombinant MIF (rMIF)(10 ng/ml). rMIF was a generous gift from Dr. Robert Mitchell at the University of Louisville and was prokaryotically expressed, purified, and refolded as described previously[57]. Cells were cultured in DMEM and incubated at 37 °C with 5% CO$_2$ for 5–6 h in the presence of 1X brefeldin A (Biolegend). The cells were then harvested using a cell scraper, washed, pelleted, and then stained for intracellular cytokine expression.

**In vitro training and restimulation assay.** Peritoneal macrophages or sorted CD11b + cells from a mouse pancreas were plated in a 24 well plate for 2 h at 37 °C and 5% CO$_2$ to allow for the attachment of cells to the plates, after which the floating cells were gently aspirated. The attached cells were gently washed with sterile PBS and then constituted in 1 mL of DMEM constituted of 10% FBS 1% penicillin/streptomycin. For the initial training of cells, 25 μg/ml of particulate WGP, 25 μg/ml of 3 μm polystyrene microparticle beads (Sigma-Aldrich), or 100 uL of PBS were added to the appropriate well and incubated for 24 h. After 24 h, wells were gently washed to remove the initial stimulus and fresh DMEM was added and cells were incubated with 5% CO$_2$ at 37 °C for 7 days. After 7 days, the media was aspirated, replaced with fresh media, and cells were restimulated with LPS (100 ng/mL), the supernatant of KPC or Pan02 cells (40%), or PBS as a control. Twenty-four hours after stimulation, the supernatants were harvested and used in an ELISA for TNF-α and IL-6.

**Tumor-conditioned media.** About 1 × 10$^6$ KPC or Pan02 cells were cultured in a six-well plate in 4 mL of complete DMEM and at 37 °C and 5% CO$_2$. After 3 days the supernatants were harvested and stored at −80 °C in aliquots for use as a tumor-conditioned medium. As a control, the pancreas of a C57BL/6 mouse was processed into a single cell suspension and 1 × 10$^6$ of these cells were cultured in a six-well plate in 4 mL of complete DMEM and at 37 °C and 5% CO$_2$, and supernatants were also collected after 3 days. As a control, the pancreas of naïve mice were processed into a single cell suspension and plated in a six-well plate for 1 day. Non-adherent cells were washed away and these cells were then cultured for 3 days and the supernatant was harvested.

**Acquisition of peritoneal macrophages.** Mice were euthanized and 5 mL of sterile RPMI was injected into the peritoneum using a 25-gauge syringe. The abdomen was massaged to liberate the peritoneal macrophages. A small incision was made and a transfer pipette was used to remove the suspension. The peritoneal cavity was then washed several times with cold RPMI and cells were pelleted at 458 × g.

**Flow cytometry.** Single-cell suspensions in PBS with 1% FBS were blocked with murine Fc Block (anti-CD16/CD32) at 4 °C for 15 min. Fluorochrome labeled antibodies for viability (APC/Cy7) and to surface markers CD45, CD11b, F4/80, Ly6G, Ly6C, MHCII, CD3, CD4, CD8, NK1.1, Ki67 (Biolegend), and CCR2 (R&D Systems) were used. The information related to all flow Abs is summarized in Supplementary Table 3. After 30 min of incubating in the dark at 4 °C, cells were washed with cold PBS and filtered through a 40 μm mesh filter. The samples were acquired using a FACSCanto II cytometer (BD Biosciences) and analyzed using FlowJo software (Tree Star, Ashland, OR). FACSDiva software version 6 was used in FACSCanto to acquire all data.

**Intracellular staining for expression of cytokines.** Following stimulation, cells were stained for the desired surface markers as described above. The cells were then washed with cold PBS, and 500 μl of fixation buffer (Biolegend) was added to the tubes, briefly vortexed, and incubated in the dark at room temperature for 20 min. About 1 ml of Permeabilization buffer (Biolegend) was then added and samples were centrifuged at 458 × g for 5 min at 4 °C followed by one more wash using 1 ml of permeabilization buffer. Cells were resuspended in 200 μl of permeabilization buffer and cells were stained with antibodies against TNF-α, Ki67, Granzyme-B, IL-12, IL-6, and IFNγ or the respective isotype control overnight at 4 °C. Cells were then washed, filtered, and data were acquired using a flow cytometer.

**FACS isolation of CD11b+macrophages for in vitro training.** The pancreas was processed into a single cell suspension as described above. Cells were washed with

1 mL of PBS, incubated with Fc block for 10 min at 4 °C followed by staining with viability dye (APC-Cy7), CD45 (PerCPcy5), and CD11b + (APC) for 30 min at 4 °C. Cells were washed with PBS and resuspended in cold MACS Running Buffer (Miltenyi Biotech). Viability$^-$CD45$^+$CD11b$^+$ cells were sorted using a FACS Aria III (BD Biosciences). Cells were collected in a 50% FBS, 40% PBS, and 10% HEPES (Corning). After sorting the cells were washed with PBS and then plated for in vitro training, as described above.

**ELISA.** Supernatants from in vitro trained cells along with standards were analyzed using murine TNF-α and IL-6 ELISA kits (BioLegend). The assay was performed per the manufacturer's instructions and all conditions were performed in triplicates. An ELISA kit for rMIF (R and D systems) was also used according to the manufacturer's instructions in order to quantify rMIF in Pan02 and KPC tumor-conditioned media and the supernatant from cultured untreated pancreatic cells. To quantitate histone modifications, ELISA kits (Tri-methyl histone H3K4 quantification kit, Acetyl-histone H3K27 quantification kit, Tri-methyl histone H3K27 quantification kit, and total histone H3 quantification kit) were purchased from EpiGentek and assays were performed based on the manufacturer's instructions. Lactate was measured by L-Lactate assay Kit I (Eton Bioscience).

**Western blot analysis.** CD11b$^+$ cells from PBS injected or WGP-injected mice (24 h) were separated using CD11b microbeads (Miltenyi Biotech). Histone was extracted according to the manufacturer's instruction (Active Motif, Inc.) and protein concentration was determined using Bradford quantification. Histone proteins were separated by SDS-PAGE 15% Tris-HCl gels and transferred onto PVDF membranes (Millipore). The membrane was blocked with 5% BSA at room temperature for 1 h and incubated overnight at 4 °C with primary antibodies including tri-methyl-histone H3 (lys4) rabbit mAb, acetyl-histone H3 (lys27) XP Rabbit mAb, tri-methyl-histone H3 (Lys27) Rabbit mAb, and histone H3 XP rabbit mAb (Cell Signaling Technology), then incubated with HRP-conjugated secondary antibodies (GE Healthcare) at room temperature for 1 h. The membrane was developed with Amersham ECL Prime Western Blotting Detection Reagent (GE Healthcare) and detected through Medical Film Processor (Konica Minolta Medical & Graphic). Precision Plus Protein Kaleidoscope Prestained Protein Standards were used as a standard protein marker (Bio-Rad).

**qRT-PCR.** After CD11b + cells had been isolated from the mouse pancreas in WGP-treated and untreated mice, cells were saved in TRIzol. RNAs were isolated and reverse transcribed using the TaqMan Reverse Transcription Reagents (qRT-PCR) amplification using the Bio-Rad MyiQ single color RT-PCR detection system. Briefly, cDNA was amplified in a 25 uL reaction mixture consisting of SYBR Green PCR super-mix (Bio-Rad), 100 ng of complementary DNA template, and selected primers (200 nM) using the recommended cycling conditions.

**H+E.** Seven days following injection with PBS/microparticle beads or WGP, the pancreata were harvested and fixed in 4% formalin for 1 week followed by embedding in paraffin according to standard procedures. Paraffin-embedded tissues were cut into 5 mm thick sections and stained with hematoxylin and eosin (H & E) for morphological analysis.

**Serum amylase measurement.** Murine serum amylase was measured using the Amylase Activity Assay Kit (Millipore Sigma) and was used according to the manufacturer's instructions. In short, mice were injected with WGP and 7 days later, blood was collected from mice using a retro-bulbar bleeding technique. These samples were used to assay for serum amylase.

**In vivo T-cell depletion.** T-cells were depleted using an anti-CD4 mAb alone, anti-CD8 mAb alone, or anti-CD4 and anti-CD8 mAbs together. Antibodies were made in-house. In this depletion procedure, mice were injected IP with WGP on day 1 and were also injected IP with 200 μg of the mAbs at day 1 and day 4 during the treatment period. Mice were euthanized on day 7. (CD4 clone GK1.5, CD8 clone 53-6.72). Depletion efficiency was confirmed on day 7.

**In vivo NK cell depletion.** NK cells were depleted through the intraperitoneal injection of 100 μg of PK136 mAb (Produced in the laboratory of Dr. Jun Yan at the University of Louisville) on days −1 and 5 during the treatment period. WGP was injected at day 0 and on day 7 animals were euthanized and the depletion efficiency of NK cells was assessed by staining pancreatic tissues for NK1.1.

**Ly6G depletion.** Neutrophils were depleted by injecting 300 μg of anti-Ly6G mAb (Bio X Cell) or isotype control Rat IgG2a (Bio X Cell) at days −1, 2, and 6 during the course of treatment. About 1 mg of WGP was injected IP on day 1. Mice were euthanized on day 7 and the pancreas was assessed for efficiency of depletion. In the tumor, model mice were injected with 300 μg of anti-Ly6G mAb or isotype control Rat IgG2a on days −2, 4, 10, and 16. Mice were given WGP on day 0 and were implanted with orthotopic KPC pancreatic tumors on day 7. Mice were

euthanized on day 21 and pancreatic tissues were stained with Ly6G to assess granulocyte depletion efficiency at that time.

**CyTOF mass cytometry sample preparation.** Mass cytometry antibodies were either purchased from Fluidigm or were created in-house by conjugating commercially available purified antibodies to the appropriate metal isotope using the MaxPar X8 Polymer or MCP9 Polymer kits (Fluidigm). Pancreatic samples from three PBS and three 7-day WGP mice were processed into a single cell solution and ex vivo stimulation was performed as described above. Cells were gently scraped from the plates using a sterile cell scraper, washed with PBS, and placed into a sterile culture tube. About $2 \times 10^6$ cells per sample were used. Cells were first stained for viability with 5 uM cisplatin (Fluidigm) in serum-free RPMI 1640 for 5 min at RT. Cells were then washed with RPMI 1640 containing 10% FBS for 5 min at $300 \times g$. Cells were stained with the surface antibodies for 30 min at RT and washed twice with Maxpar Cell staining buffer (Fluidigm). For staining on intracellular cytokines, cells were then fixed with 1 mL of 1X Maxpar Fix I buffer for 30 min at RT and then washed twice with 2 mL of 1X Maxpar Perm-S buffer for 5 min at $800 \times g$. The cytoplasmic/secreted antibody cocktail was then added and incubated with the cells for 30 min at RT. Following incubation, cells were washed with 1 mL of 1X Maxpar Perm-S buffer for 5 min at $800 \times g$ and gently blotted to remove all liquid from the tube. In order to stain for nuclear antigens, cells were then suspended in 1 mL of 1X Maxpar nuclear antigen staining buffer for 30 min at RT. The nuclear antigen antibody cocktail was then added and incubated for 30 min at RT. Cells were washed twice for 5 min at $800 \times g$ with 2 mL of Nuclear Antigen Staining Permeability buffer. Finally, cells were fixed with 1.6% formaldehyde for 10 min at RT, then incubated overnight in 125 nM of Intercalator-Iridium (Fluidigm) at 4 °C.

**CyTOF data acquisition.** Once cells were ready for acquisition, samples were washed twice with Cell Staining Buffer (Fluidigm) and kept on ice while awaiting acquisition. Directly prior to the acquisition, cells were suspended in a 1:9 solution of Cell Acquisition Solution: EQ 4 element calibration beads (Fluidigm). A Helios CyTOF system was used, and following proper startup and tuning procedures, samples were run at a rate of less than or equal to 500 events/second up to 300,000 events. Using the CyTOF software, FCS files were normalized into.fcs files and these files were then ready for data analysis.

**CyTOF data analysis.** CyTOF data was analyzed using FlowJo, the CytoBank software package[58], and the CyTOF workflow[59] which includes a suite of packages available in R (r-project.org)[60–63]. For analysis conducted within the CyTOF workflow, FlowJo workspace files exported from flow Workspace and CytoML were used[61].

**RNA sequencing: RNA extraction and isolation.** Seven days following administration of PBS or WGP IP, pancreata were harvested, processed into a single cell suspension, stained for viability, CD45, and CD11b, and sorted using a FACS Aria III. Samples were prepared in triplicate for each experimental group. Once these myeloid cells were isolated, cells were washed 2x with ice-cold PBS and then lysed with Trizol (Invitrogen). RNAs were extracted using a QIAGEN RNAeasy Kit (QIAGEN). The isolated RNA was checked for integrity using the Agilent Bioanalyzer 2100 system (Agilent Technologies, Santa Clara, CA) and quantified using a Qubit fluorometric assay (Thermo Fisher Scientific, Waltham, MA). Poly-A enriched mRNA-Seq libraries were prepared following the Universal Plus mRNA-Seq kit standard protocol (Tecan Genomics, Redwood City, CA) using 10 ng of total RNA. All samples were ligated with Illumina adapters and individually barcoded. The absence of adapter dimers and a consistent library size of ~300 bp was confirmed using the Agilent Bioanalyzer 2100. The library concentration and sequencing behavior was assessed in relation to a standardized spike-in of PhiX using a Nano MiSeq sequencing flow cell from Illumina. 1.8 pM of the pooled libraries with 1% PhiX spike-in was loaded on one NextSeq 500/550 75 cycles High Output Kit v2 sequencing flow cell and sequenced on the Illumina NextSeq 500 sequencer targeting 60 M 1x75 bp reads per sample.

**RNA sequencing.** Libraries were prepared using the Universal Plus mRNA-seq kit with NuQuant® library quantification (NuGen). The six samples were spread across four sequencing lines in one run. The 24 single-end raw sequencing files (.fastq)[64] were downloaded from Illumina's BaseSpace[65] (https://basespace.illumina.com/) onto the KBRIN server for analysis. Quality control (QC) of the raw sequence data was performed using FastQC (version 0.10.1)[66]. The sequences were aligned to the mm10 mouse reference genome using STAR (version 2.6)[67], generating alignment files in bam format. Differential expression of ENSEMBL protein-coding transcripts was performed using DESeq2[68,69]. Raw counts were obtained from the STAR aligned bam format files using HTSeq (version 0.10.0)[70]. The raw counts were normalized using the Relative Log Expression (RLE) method and then filtered to exclude genes with fewer than ten counts across the samples[71]. RNA-seq data were deposited with GEO accession GSE187464.

**RNA sequencing: gene set enrichment analysis**. Gene set enrichment analysis (GSEA) was used to further characterize the biology of the genes comprising the WGP vs PBS conditions and their differences[72]. Gene sets were obtained from the Molecular Signatures Database (MSigDB) for Gene Ontology (GO) biological processes and Reactome pathways. For each gene set, all tested gene locations in the comparison of WGP vs PBS-treated control mice were sorted from highest to lowest significance using p values. This approach allows highly significant up- and downregulated genes to be included within each gene set, an approach that more accurately reflects the conditions in a biological pathway. For this analysis, a table of the enriched sets is followed by an enrichment plot displaying the profile of the enrichment score (ES) and the position of gene set members on the rank-ordered list.

**Single-cell sequencing: isolation of single cells and RNA sequencing**. Live CD45 + cells were sorted from mouse pancreata, washed, and resuspended in 1x PBS (calcium and magnesium-free) containing 0.04% BSA. Single cells were captured and barcoded cDNA libraries were constructed using the Chromium Next GEM Single-Cell 3′ Reagent Kit (v3.1, 10X Genomics) and the Chromium Controller, according to the manufacturer's instructions. Libraries were pooled and sequenced using a 28 bp × 8 bp × 125 bp configuration for read1 × i7 index x read2 on the Illumina NextSeq 500 with the NextSeq 500/550 150 cycle High Output Kit v2.5 (20024907).

**Since cell sequencing: gene expression profiling**. Bcl files were demultiplexed into fastq files using the CellRanger software (10X Genomics, v3.1.0). The total number of sequenced reads was 506,913,062. The reads were of good quality as determined by FastQC[66]. Gene counts were measured using CellRanger "count", utilizing the cellranger-mm10-3.0.0 reference genome for mouse. A counts matrix was generated for each individual sample and one aggregated sample with the expected number of cells set at 5000.

The raw count data determined by CellRanger was used as input to a custom analysis pipeline in R which uses a variety of single-cell analysis tools based on Seurat. The knee plot (Supplementary Fig. 7b) displays a graph showing the ranked UMI counts for each cell barcode for data aggregated across the three groups. Cells above the inflection point represent possible doublets while those below the knee represent background cells. Cell quality control measures were analyzed using Seurat v3[73], and cell barcodes with the following characteristics were removed from the analysis: low counts (possible background cells) with an FDR cutoff of 0.01 from the DropletUtils[74] function "emptyDrops", high counts (possible doublet cells) with more counts than the knee plot inflection point, mitochondrial content greater than 30% and ribosomal content greater than 40%. Gene (features) were further filtered to remove retired gene identifiers and genes that were not expressed in at least two cells. See Supplementary Fig. 7a, b for the initial and filtered number of cell barcodes and genes.

The expression data were normalized using SCTransform[75] where cell cycle genes, ribosomal content, and mitochondrial content were regressed. The cells were then clustered and dimension reduction was performed using UMAP[76]. Initial cluster names were assigned using a modified GSVA[77,78] enrichment score technique. For each of these clusters, the top marker genes were identified. Differentially expressed genes comparing each cluster to every other cluster (all pairwise comparisons) was determined using Seurat[73] and MAST[79]. scRNA-seq data were deposited with GEO accession GSE187464.

**Identification of clusters generated with scRNA-Seq**. Non myeloid-derived clusters were classified generally as B-cells (*MS4A1*), plasma cells (*SDC1*), CD8 + T-cells (*CD3e, C8a*), CD4 + T-cells (*CD3e, CD4*),T-regulatory cells (T-regs) (*CD3e, CD4, FoxP3*), γδ T-cells (*CD3e, TRGC1, TRGC2,IL7Ra*) and type 2 innate immune cells (ILC2s) (*Alox5, KLRG1, Ly6a, Pparg,GATA3, IL-5, IL-13, ICOS and Rxrg*)[80–82]. Neutrophils were identified through *MMP9, Csf3r, S100A8, S100A8*, and *ADAM8* expression, though may be underrepresented in these analyses due to their low RNA content and high levels of intrinsic RNases[83]. Conventional dendritic cells (cDCs) were identified through *ITGAX* and *ITGAE* expression and plasmacytoid DCs (pDCs) were identified by *ITGAX* and *Siglech*.

**Classification of myeloid clusters from scRNA-Seq**. Cluster 5 expressed $ITGAM^{Int}ADGRE1^{Hi}Lyz2^{Hi}H2\text{-}Ab1^{Hi}Ly6C2^{-}$ and did not express *TNFAIP2*, indicating that these cells are resident macrophages. Cluster10 expressed $ITGAM\text{-}^{Hi}ADGRE1^{Hi}Lyz2^{Hi} H2\text{-}Ab1^{Int} Ly6C2^{-}$. Cluster 3 and 4 expressed $ITGAM^{Int} ADGRE1^{Int} Lyz2^{Hi} H2\text{-}Ab1^{Int} Ly6C2^{Hi}$, which suggests that both are subsets of infiltrating monocytes/macrophages, though the enhanced inflammatory genes expressed in cluster 3 were used to identify cluster 3 as an inflammatory infiltrating monocyte/macrophage.

**Phagocytosis assays**. The pancreas of PBS/microparticle injected mice and in vivo WGP-trained mice were harvested 7 days after injection and processed as described previously into a single cell suspension. About $2 \times 10^6$ were washed with HEPES dilutes 50x in RPMI 1640 and then incubated in 100 µL of this solution for 1 h at 37 °C in order to activate the cells. The Invitrogen pHrodo™ Green S.

*aureus* BioParticles™ Phagocytosis Kit for Flow cytometry (Thermo Fisher Scientific) was used according to the manufacturer's instructions. About 100 µL of the reconstituted particles or $1 \times 10^6$ GFP + KPC tumor cells were added to the activated pancreatic cells and incubated for 1 h at 37 °C. Samples were gently vortexed every 15 min. The reaction was stopped by adding 1 mL of cold PBS. Samples were incubated for 10 min at 4 °C, stained for viability, CD45, CD11b, and F4/80 for 30 min at 4 °C, and then analyzed using a BD FACSCanto. For analysis, after gating on live cells and CD45, cells that fluoresced in the FITC channel were determined to be phagocytic.

**Cytotoxicity assay**. The pancreas from WGP and PBS-treated mice were harvested and the CD11b + populations were isolated using magnetic CD11b + MicroBeads (Miltenyi Biotec) and an autoMACS Pro Separator (Miltenyi Biotec). Purified CD11b + cells were then washed and counted, and these were plated at a ratio of 1:20 tumor:effector cells in a 96 well plate. All experimental samples were run in triplicate. After plating the CD11b + cells, the ROS inhibitors *N*-acetyl-ʟ-cysteine (NAC) (1 mM, Sigma-Aldrich), Trolox (5 µM, Sigma-Aldrich), or DFO (50 µM, European Pharmacopoeia) were added to one set of PBS and WGP derived cells for 1 h before the addition of 5000 luciferase-expressing KPC + pancreatic tumor cells to all wells. After 24 h, of co-culture, the plates were centrifuged and 20 µL of the supernatant was mixed with 100 µL of the Luciferase Assay Reagent (Promega). Luciferase activity measured in the supernatant correlated with tumor cells that had been killed by the effector cells and was measured using a luminometer (Femtomaster FB 12, Zylux Corporation). The spontaneous luciferase signal from plated tumor cells was subtracted from the measurement of the supernatant. Luciferase values are represented as Relative Light Units (RLUs).

**In vivo tumor models of pancreatic cancer**. A KPC cell line on a C57BL/6 background derived from the $LSL\text{-}Kras^{G12D/+}$; $LSL\text{-}Trp53^{R172H/+}$; *Pdx1-Cre* (KPC) mouse model was purchased from Ximbio. These and Pan02 cells which were a generous gift from Dr. Yong Lu at Wake Forest University were used in an orthotopic model of pancreatic cancer. A KPC line transfected with GFP and luciferase ($KPC^{GFP+Luc+}$) were also a generous contribution from Dr. Michael Dwinell at the Medical College of Wisconsin. These cells were used exclusively in albino C57BL/6 mice. For tumor implantation, mice were anesthetized using isoflurane, and the abdomen of the mice were prepped with betadine and draped in a sterile fashion. A 2 cm midline laparotomy was performed using an aseptic technique with sterile instruments. Following laparotomy, the pancreas and spleen were externalized. Tumor cells were suspended in ice-cold PBS and mixed in a 1:1 ratio with basement membrane matrix Matrigel (Corning). $0.1 \times 10^6$ tumor cells in 50 uL of the PBS-matrigel solution were injected into the tail of the pancreas using a 30-gauge insulin syringe. The formation of a small bubble indicated successful implantation. The peritoneum was closed using coated polyglycolic acid braided absorbable 5/0 suture and the skin was closed using silk braided nonabsorbable 5/0 suture. (CP Medical). Buprenorphine was administered for pain management up to 72 h following surgery and mice were monitored.

**In vivo imaging**. Mice implanted orthotopically with GFP + Luciferase+ KPC tumor cells were injected IP with 150 mg/kg of body weight at 100 µL of XenoLight D-Luciferin-K + Salt Bioluminescent Substrate (PerkinElmer). After 10 min, mice were anesthetized with isoflurane and placed inside of a Biospace Lab Photon Imager, which is a dedicated low light level in vivo optical modality for bioluminescent and fluorescent imaging. Images of mice were taken and used to measure tumor size and growth.

**Admixture tumor model**. Mice were trained with 1 mg of WGP and 7 days later the CD11b+CCR2+ and CD11b+CCR2− populations were sorted and mixed 1:1with KPC tumor cells $1 \times 10^5$ tumor cells. Cells were implanted orthotopically into WT mice and 3 weeks later mice were euthanized and tumor size was assessed.

**Combination therapy with anti-PD-L1 mAb**. C57BL/6 mice were treated with WGP on day −7 and implanted with KPC orthotopic pancreatic tumors on day 0. Mice were treated with 200 µg of anti-mouse anti-PD-L1 mAb (Bio X Cell) or Rat IgG2b isotype control (Bio X Cell) on days 3, 7, and 11. Mice were then monitored for survival.

**WGP as a treatment**. C57BL/6 mice were implanted with KPC orthotopic pancreatic tumors at day 0 and on days 4 and 11 mice were given 1 mg of IP WGP or PBS. Mice were then monitored for survival.

**Statistical analysis**. All results were repeated at least three independent times to verify unless otherwise specified. Representative results are shown. Results are represented as mean ± SEM. Data were analyzed using a two-tailed Student's *t*-test or Mann–Whitney *U*-test. Multiple-group comparisons were performed using a one-way or two-way ANOVA followed by Tukey's multiple comparisons test. Correlation analyses were performed using the Pearson correlation coefficient (normal distribution). Statistical significance was set at $p < 0.05$. All statistical

analyses were performed using GraphPad Prism Software Version 9 (GraphPad Inc., La Jolla, CA).

**Reporting summary**. Further information on research design is available in the Nature Research Reporting Summary linked to this article.

## Data availability

Source data are provided with this paper. RNASeq data is available under GEO accession GSE187464, and scRNA-Seq data is available under GEO accession GSE187464 Source data are provided with this paper.

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

## Acknowledgements

The authors wish to thank Dr. Wolfgang Zacharias, Dr. Mei Zhang, and Sabine Waigel from the Brown Cancer Center Genomics Facility for their help in RNA-Seq and scRNA-Seq. We would also like to thank Dr. Clint Geller for proofreading the manuscript. This work was supported by the NIH R01CA213990 and R01AI128818 (J.Y.). C.D. and C.T.W. were supported in part by the NIH P20GM135004. Part of this work was performed with the assistance of the UofL Genomics Facility, which is supported by NIH P20GM103436 (KY IDeA Networks of Biomedical Research Excellence), the Brown Cancer Center, and user fees. CyTOF was performed in the Functional Immunomics Core supported by NIH P20GM135004 (Jun Yan/Jason Chesney, MPI). Sequencing and bioinformatics support for this work was provided by NIH P20GM103436 (Nigel Cooper, PI) and NIH P20GM106396 (Donald Miller, PI). The contents of this work are solely the responsibility of the authors and do not represent the official views of the NIH or the National Institute for General Medical Sciences (NIGMS).

## Author contributions

A.E.G. designed and performed the experiments, analyzed data, interpreted results, and wrote the manuscript. R.S., C.D., and H.G. contributed to experimental design, data acquisition, and data analysis. M.R.W., X.H., and M.Z. assisted in data acquisition and preparation of materials. K.A. and E.C.R. contributed to the computational analysis and interpretation of RNA-Seq analyses and J.H.C. and E.C.R. contributed to the computational analysis and interpretation of ScRNA-Seq analyses. D.T and C.T.W. contributed to the analysis of CyTOF data. R.A.M., H.-g.Z., and Y.L. contributed materials and provided guidance for the project. R.A.M. performed a critical review. J.Y. directed experimental design and the overall study, interpreted data, supervised research, and assisted in writing and review of the manuscript.

## Competing interests

The authors declare no competing interests.
