## [Peer Review File · Nature Communications]

The Induction of Peripheral Trained Immunity in the Pancreas Incites Anti-tumor Activity to Control Pancreatic Cancer ProgressionREVIEWER COMMENTS

Reviewer #1 (Remarks to the Author):

This is an interesting study investigating the impact of the induction of trained immunity on the outcome of pancreatic cancer. The authors report the unexpected property of beta-glucan to traffic to the pancreas, where it induces functional reprogramming of myeloid cells. This leads to enhanced phagocytosis and ROS production against tumor cells, and improvement of the outcome of the experimental animals. This beneficial effect of trained immunity induction is further amplified by checkpoint inhibitors. The study is very relevant, novel, and the experiments are well-performed, containing all the important components to support the conclusions.

Comments:

1. The authors nicely show the induction of a trained immunity phenotype by the innate immune cell infiltrating the pancreas after injection of beta-glucan i.p. While they measure a number of relevant cytokines to characterize this phenotype, other markers of trained immunity such as IL-1b and lactate release can also be considered.
2. The authors show very nicely the transcriptional changes induced by beta-glucan in the myeloid cells. Are these changes also accompanied by increased chromatin accessibility, as shown in other models of trained immunity?
3. Can the authors discuss whether differences in effects of particulate vs soluble beta-glucan could be expected in this model.
4. Can the authors speculate what are the chances that a soluble beta-glucan injected i.v. could also accumulate in the pancreas?
5. It is unclear from the description of the methods if the crucial experiments were repeated and are reproducible.

Reviewer #2 (Remarks to the Author):

This is a very novel and exciting manuscript. The authors show that innate immunity can be trained in the pancreatic cancer microenvironment. This work has important scientific implications to the greater field of cancer immunology as well as to translational therapeutics. The experiments are well-designed and the manuscript is very clearly written. All my comments are minor.

- Figure 1a: <1%, 4%, 6%, 1%, 18%.....whole numbers are more readable

- Figure 1a: How do u distinguish peritoneal cavity from pancreas b/c they are presumably on pancreas, rather than in pancreas? Perhaps this can be discussed.

-Figure 1a: Consider quatifying with bar graph

-Figure 1b,c: No stats

- Figure 2 and S1: Absolute numbers (supp data) may be more important than % (main figure). Can consider flipping location

- Figure 5F: How does this work? Can authors reference paper showing direct myeloid toxicity?

- Figure 6: Authors may want to move this figure earlier in paper as it is so important and takes a while to get to (optional).

- Figure 6: Why did the authors focus on CCR2 and not MIF? Please discuss

-Figure 7: How does antiPDF-L1 combo treatment work. Are u combining adaptive and innate

immunity or is it all innate? This should be discussed.

Reviewer #3 (Remarks to the Author):

The authors did a great job characterizing β -glucan trafficking selectively to pancreas. They perform a series of comprehensive in-vitro and in-vivo experiments to dissect the mechanisms behind the phenotype. Authors use single cell RNA sequencing approach to explore the mechanism and found that the trafficking of β -glucan results in CCR2 dependent influx of macrophages/monocytes to pancreas and displays features of trained immunity. It is interesting that the trained cells can be activated upon exposure to tumor or tumor derived factors. Additionally, they also show enhanced phagocytosis and ROS mediated killing of PDAC tumor cells. Overall, novelty of the data presented in the manuscript and the rigorous use of both positive and negative controls in the experiments performed warrants a publication. However, if authors could answer few questions and correct the typos and figure annotation mentioned below, it would help readers understand the topic better.

Major Comments:

1. Badgley and Kremer et.al., Science 2020, <https://science.sciencemag.org/content/368/6486/85/tab-pdf> showed that PDAC tumor is sensitive to ROS and lipid oxidation. In figure 5F authors show that treatment of WGP and NAC abrogates the cytotoxicity of WGP trained cells. Do the authors know if this is dependent on lipid peroxidation? Have they tried using other ROS scavengers such as Trolox or DFO to scavenge the cellular ROS?
2. It is understandable that authors focus primarily on macrophage/monocyte as of result of increased Ccr2/Ccl2 levels, however, there is marked depletion in ILC2 population 7 days after WGP treatment (Figure 3A, 3D). ILC2 is known to infiltrate PDAC and activate tissue-specific immunity, if authors could comment on how ILC2 expansion (cluster 11) at day 3 and depletion of same cluster 11 at day 7 post WGP treatment is related?
3. In figure 3G, authors show that anti-inflammatory clusters 5 and 10 express high levels of Arg1. In figure 5E, following WGP treatment, there is increased Arg2 and Slc7a2, suggesting reliance in Arginine catabolism for the trained cells, does arginine starvation of trained cells have similar effect as NAC treatment? Additionally, what is the status of immuno- modulatory cytokines such as IL4, IL5, IL13, IL33 after ROS inhibition and Arginine starvation?
4. Except the differential pathway like phagocytosis and ROS related pathways, what are the top pathways in GESA analysis? Which may also contribute to the antitumor effects of IP WGP.
5. what is the mechanism of the increase in the proportion of CD8+ T-cells present within CCR2+ admixed tumors? This will be important for optimization of WGP and anti-PD-L1 combination.

Minor Comments:

1. Figure 2C, it would be easier for the reader if the pie chart was annotated with percentage
2. Figure 2I, please increase the font size of the gene names
3. Figure 2J, please add the direction of comparison e.g., PBS vs WGP
4. Figure 3A, the cluster labels are too small, please increase the font size
5. Figure 3D, there is typo in cluster label, in figure 3A, the red cluster is labelled "0", but in panel D, its labelled "4", also cluster labels of figure 3D is very small.
6. Figure 4E, please add the direction of comparison e.g., PBS vs WGP

Dear Reviewers:

The enclosed manuscript entitled, “**The Induction of Peripheral Trained Immunity in the Pancreas Incites Antitumor Activity to Control Pancreatic Cancer Progression**” (NCOMMS-21-12966-T) has been revised according to the insightful suggestions and comments from Reviewers and is hereby submitted for the possible publication in *Nature Communications*. A point-by-point reply to the Reviewers’ comments follows. Please note, the revisions/additions are marked by **yellow font color highlighting** in the manuscript, while deletions are not shown in the revised manuscript.

Point-by-Point Reply to the Reviewer’s Comments

Reviewer#1:

1. The authors nicely show the induction of a trained immunity phenotype by the innate immune cell infiltrating the pancreas after injection of beta-glucan i.p. While they measure a number of relevant cytokines to characterize this phenotype, other markers of trained immunity such as IL-1b and lactate release can also be considered.

Response: In order to address whether lactate or IL-1 β release can be used as markers of trained immunity, CD11b⁺ cells were microbeads sorted from pancreas of PBS, 3-day and 7-day WGP trained mice. These cells were re-exposed to LPS, and levels of lactate and IL-1 β release were measured by L-lactate assay kit or IL-1 β ELISA kit, respectively. As shown in the following figure panel A, lactate production was significantly increased upon WGP training (3 days and 7 days) compared to untrained CD11b⁺ cells.

However, IL-1 β production was overall quite low (data not shown). We had previously observed in our RNA-Seq data that *IL-1b* mRNA level was significantly increased in WGP trained myeloid cells (panel B).

Together this may indicate that while IL-1 β is a marker of the trained response, IL-1 β protein level is relatively low as compared to other markers of trained immunity such as TNF- α , making TNF- α a better surrogate marker. We have included these data in the supplemental Figure 1G.

2. The authors show very nicely the transcriptional changes induced by beta-glucan in the myeloid cells. Are these changes also accompanied by increased chromatin accessibility, as shown in other models of trained immunity?

Response: In this revised submission, we examined the epigenetic markers of trained immunity. We focused on the histone modifications since previous studies have shown that trained immunity is associated with histone modifications, particularly H3K4me3, H3K27Ac, and H3K27me3. Two assays were used to assess the histone modifications. One was done with Western blot analysis and another was using ELISA to quantitatively measure these epigenetic markers. As shown in the right figure, upon WGP training, pancreas CD11b⁺

innate immune cells exhibited increased expression of epigenetic markers H3K4me3 and H3K27Ac while H3K27me3 was slightly lower compared to untrained cells. This was revealed by both western blot analysis (Panel A) and ELISA quantitative assay (Panel B). These data are also consistent with previous reports that trained immunity is associated with epigenetic reprogramming. Taken together, this additional data further supports the notion that WGP-induced trained immunity in pancreas is also associated with epigenetic modifications. We have added this data as Figure 2I in the revised manuscript and further discussed it (page 10).

3. Can the authors discuss whether differences in effects of particulate vs soluble beta-glucan could be expected in this model.

Response: Thanks for the suggestion. In this revised submission, we have discussed the differences in effects of particulate vs soluble beta-glucan on pancreas trained immunity phenotype. This has been included in the discussion on page 23-24.

4. Can the authors speculate what are the chances that a soluble beta-glucan injected i.v. could also accumulate in the pancreas?

Response: Thanks for the suggestion. This has also been included in the discussion on page 23-24.

5. It is unclear from the description of the methods if the crucial experiments were repeated and are reproducible.

Response: Sorry for this oversight. All experiments were performed in triplicate unless otherwise specified. Representative images are shown. If experimental repeats could not be combined due to small baseline changes but the overall trend remained consistent and statistically significant, representative images are shown. A clarification has been added in the statistics section on page 48.

Reviewer #2 (Remarks to the Author):

This is a very novel and exciting manuscript. The authors show that innate immunity can be trained in the pancreatic cancer microenvironment. This work has important scientific implications to the greater field of cancer immunology as well as to translational therapeutics. The experiments are well-designed and the manuscript is very clearly written. All my comments are minor.

Response: Thanks for the positive comments on our manuscript.

- Figure 1a: <1%, 4%, 6%, 1%, 18%.....whole numbers are more readable

Response: We understand and appreciate your comment on this issue. However, we also want to indicate that there is a difference between <1% and .0184% and want this to be clear in the manuscript. As such if approved by you, we would like to leave the percentages as they stand.

- Figure 1a: How do u distinguish peritoneal cavity from pancreas b/c they are presumably on pancreas, rather than in pancreas? Perhaps this can be discussed.

Response: In order to assess peritoneal cavity, PBS was injected into the peritoneum of mice and these cells were then carefully extracted from the abdominal cavity with minimal perturbation to the pancreas. By contrast, when pancreatic cells were assessed, the pancreas was washed with ice-cold PBS several times before processing. This would have presumably washed away cells loosely associated with the surface from the peritoneum. It is likely, however, that there is a dynamic flux of cells between the pancreas and the peritoneum making it difficult to decipher cells from the peritoneal cavity vs the pancreas. This is discussed on page 23 of the discussion. This process of carefully washing the organs before processing was also added on page 6.

-Figure 1a: Consider quantifying with bar graph

Response: Thanks for the suggestion. The bar graph has been added in Figure 1a.

-Figure 1b,c: No stats

Response: This has been updated in 1b and 1c. For 1c, stats are included that show significance as related to the pancreas as all other comparisons are not significant. This was updated in the figure and figure legend.

- Figure 2 and S1: Absolute numbers (supp data) may be more important than % (main figure). Can consider flipping location

Response: Thanks for the suggestion. In this revised submission, we included absolute numbers in the main figure (Figure 2) and % in the supplemental figure.

- Figure 5F: How does this work? Can authors reference paper showing direct myeloid toxicity?

Response: It has been shown in many papers that myeloid cells, including macrophages and monocytes, within the TME often exert anti-tumor functionalities by phagocytosing tumor cells. We show that WGP training upregulates phagocytosis and show that ROS play a role in the myeloid-mediated cell death. Several papers that discuss direct myeloid toxicity to tumor cells have been cited in the discussion (Page 26).

- Figure 6: Authors may want to move this figure earlier in paper as it is so important and takes a while to get to (optional).

Response: We appreciate the reviewer's suggestion. While we certainly understand this recommendation, we believe that the flow of the paper makes more sense by defining the mechanism of trafficking and exploring the potential anti-tumor effects of trained immunity and then applying these to a model of cancer.

- Figure 6: Why did the authors focus on CCR2 and not MIF? Please discuss

Response: We chose to focus on CCR2 because as we were conducting our phenotypic studies of the types of cells that were present in the pancreas following WGP injection, we noticed that a majority of the new cells expressed CCR2. The expression was so striking that we believed CCR2 signaling could be responsible for the influx of these cells into the pancreas. Further studies showed that we were correct in that CCR2 signaling does drive the influx of cells into the pancreas. MIF on the other hand is involved in re-activating the trained innate immune cells that enter the pancreas, but is not responsible for the trafficking of these cells like CCR2 is. MIF was found in high levels in the supernatant of pancreatic cancer cells, but was not found in appreciable levels in a naïve pancreas. We believe MIF acts as one of several stimuli that controls the re-activation of the trained innate immune response. While this finding is novel in this context, the finding that CCR2 drives influx of trained innate immune cells into the pancreas is more integral to the novel mechanism of immune trafficking to the pancreas and has not been shown before. As such, we chose to focus on CCR2. Future studies could certainly focus on the role of MIF in reactivating the trained immune response.

-Figure 7: How does antiPDL-L1 combo treatment work. Are u combining adaptive and innate immunity or is it all innate? This should be discussed.

Response: We did initially partially address where in the discussion (Page 27) we stated that “We also note high expression of MHC II on these trained monocyte/macrophages, which likely plays a role in tumor antigen processing and communication with T-cells to elicit adaptive immune responses against the tumor. Interestingly, while we confirmed that these antitumor mechanisms do not depend on adaptive immune responses, we did observe a significant increase in CD8⁺ T-cells present within CCR2⁺ admixed tumors.”

We have added additional text to extrapolate more based on this comment and comments from Reviewer #3 (point 5) on the interplay between the CCR2⁺ myeloid cells and CD8⁺ T-cell recruitment. “scRNA-Seq data showed that the CCR2⁺ monocytes/macrophages that traffic into the pancreas by day 7 have high expression of CXCL9 and CXCL10, both of which are critical in the recruitment of CD8⁺ T-cells (data not shown)^{56, 57}. Thus these CCR2⁺ cells are likely directly responsible for the recruitment of CD8⁺ T-cells to the pancreas.” From this, we imply that the mechanism of anti-tumor immunity is driven by innate immune cells, however these trained innate immune cells likely potentiate and activate the effects of anti-tumor adaptive immune cells which is further unleashed by the anti PD-L1 therapy. This is on pages 27 and 28.

Reviewer #3 (Remarks to the Author):

The authors did a great job characterizing b-glucan trafficking selectively to pancreas. They perform a series of comprehensive in-vitro and in-vivo experiments to dissect the mechanisms behind the phenotype. Authors use single cell RNA sequencing approach to explore the mechanism and found that the trafficking of b-glucan results in CCR2 dependent influx of macrophages/monocytes to pancreas and displays features of trained immunity. It is interesting that the trained cells can be activated upon exposure to tumor or tumor derived factors. Additionally, they also show enhanced phagocytosis and ROS mediated killing of PDAC tumor cells. Overall, novelty of the data presented in the manuscript and the rigorous use of both positive and negative controls in the experiments performed warrants a publication. However, if

authors could answer few questions and correct the typos and figure annotation mentioned below, it would help readers understand the topic better.

Major Comments:

1. Badgley and Kremer et.al., Science

2020, <https://science.sciencemag.org/content/368/6486/85/tab-pdf> showed that PDAC tumor is sensitive to ROS and lipid oxidation. In figure 5F authors show that treatment of WGP and NAC abrogates the cytotoxicity of WGP trained cells. Do the authors know if this is dependent on lipid peroxidation? Have they tried using other ROS scavengers such as Trolox or DFO to scavenge the cellular ROS?

Response: Thanks for the comments. ROS-induced lipid peroxidation plays a critical role in inducing cell death including apoptosis, autophagy, and ferroptosis. Although we had not previously investigated this, we used two additional ROS scavengers including Trolox and DFO to examine their effect on WGP-trained innate cells-mediated cytotoxicity against pancreatic cancer cells. For this experiment, CD11b⁺ myeloid cells from 7-day WGP trained mice were microbeads separated and plated these for a cytotoxicity assay with Luciferase⁺ KPC cells as target cells for 24 hours with experimental conditions including Trolox and DFO to inhibit lipid peroxidation. We showed that both Trolox and DFO significantly reduced the ability of WGP trained cells to kill tumor cells (right figure). This data supports that ROS-induced lipid peroxidation does play an important role, at least partly, in this myeloid-cell mediated killing of tumor cells. We have included this data in Figure 5G and further discussed this on page 18.

2. It is understandable that authors focus primarily on macrophage/monocyte as of result of increased Ccr2/Ccl2 levels, however, there is marked depletion in ILC2 population 7 days after WGP treatment (Figure 3A, 3D). ILC2 is known to infiltrate PDAC and activate tissue-specific immunity, if authors could comment on how ILC2 expansion (cluster 11) at day 3 and depletion of same cluster 11 at day 7 post WGP treatment is related?

Response: Thank you for commenting on this interesting phenomenon! This has been added to the discussion. We also thought this relative enhancement and then disappearance of these ILC2s was interesting, though did not believe it to be ultimately responsible for the anti-tumor effects observed in this model so did not investigate this further. There could be many mechanisms responsible for this relative enhancement and then disappearance, and a brief discussion of these dynamics have been added into the discussion on pages 24-25.

3. In figure 3G, authors show that anti-inflammatory clusters 5 and 10 express high levels of Arg1. In figure 5E, following WGP treatment, there is increased Arg2 and Slc7a2, suggesting reliance in Arginine catabolism for the trained cells, does arginine starvation of trained cells have similar effect as NAC treatment? Additionally, what is the status of immuno- modulatory cytokines such as IL4, IL5, IL13, IL33 after ROS inhibition and Arginine starvation?

Response: Based on this comment we performed a cytotoxicity experiment using microbeads separated CD11b⁺ cells from 7-day WGP trained mice. These cells were cultured with luciferase-expressing KPC cells in the normal complete RPMI1640 media or arginine-deprived media. Myeloid-cell cytotoxicity was significantly decreased in the setting of arginine depletion, suggesting that trained cells are at least partially reliant on arginine catabolism. Although we elected not to include this data in the revised manuscript, we will further explore this line of work in the future.

Regarding other immune-modulatory cytokines such as IL-4, IL-5, IL-13 and IL-33, they were found to not be differentially regulated by WGP training based on the RNAseq data. Because there was not a difference in the expression of these markers in the presence of PBS vs WGP training (right figure), ROS inhibition or arginine starvation would presumably not impact these results.

4. Except the differential pathway like phagocytosis and ROS related pathways, what are the top pathways in GESA analysis? Which may also contribute to the antitumor effects of IP WGP. Include this?

Response: In this revised submission, we included two supplemental tables to detail the top 20 pathways (GO pathways and KEGG pathways) in GESA analysis. Other upregulated pathways such as complement pathway and proteasome pathway may also contribute to the antitumor effects of WGP-induced trained immunity.

5. what is the mechanism of the increase in the proportion of CD8⁺ T-cells present within CCR2⁺ admixed tumors? This will be important for optimization of WGP and anti-PD-L1 combination.

Response: We believe that CD8⁺ T-cells are likely recruited to the tumor through the secretion of chemotactic factors by the CCR2⁺ cells that enter the pancreas. scRNA-seq data has shown that the CCR2⁺ clusters that enter the pancreas by day 7, specifically cluster 3, highly expresses CXCL10 and CXCL9 as shown in the following figure. CXCL9 and CXCL10 are both chemokines that are known to recruit CD8⁺ T-cells to a tumor. In line with this, RNASeq data

(supplemental Table 1) showed that antigen processing and presentation was among the top 20 enriched KEGG pathways in

WGP trained pancreatic CD11b⁺ cells. Therefore, it is possible that trained myeloid cells serve as potent antigen-presenting cells to activate CD8⁺ T cells and anti-PD-L1 could further enhance

this effect. We have included this information in the discussion (page 27-28). This is also an active research direction in the laboratory.

Minor Comments:

1. Figure 2C, it would be easier for the reader if the pie chart was annotated with percentage

- This has been fixed and % has been included

2. Figure 2I, please increase the font size of the gene names

- The font size has been increased.

3. Figure 2J, please add the direction of comparison e.g., PBS vs WGP

- In this revised submission, the original Figure 2J was moved to the supplemental figures as Fig S2A. This has been included in the figure legend for Fig S2A. “PBS was used as the control and compared to WGP”

4. Figure 3A, the cluster labels are too small, please increase the font size

- These were made larger

5. Figure 3D, there is typo in cluster label, in figure 3A, the red cluster is labelled “0”, but in panel D, its labelled “4”, also cluster labels of figure 3D is very small.

- This error has been fixed and the font has been made larger. Thank you for catching this.

6. Figure 4E, please add the direction of comparison e.g., PBS vs WGP

- This has been included in the figure legend for 4E. “PBS was used as the control and compared to WGP”

REVIEWER COMMENTS

Reviewer #1 (Remarks to the Author):

The authors responded appropriately to my comments.

Reviewer #2 (Remarks to the Author):

All my concerns have been well-addressed.

Reviewer #3 (Remarks to the Author):

My concerns have been well addressed.